# One-quarter of freshwater fauna threatened with extinction

Freshwater ecosystems are highly biodiverse[1] and important for livelihoods and economic development[2], but are under substantial stress[3]. To date, comprehensive global assessments of extinction risk have not included any speciose groups primarily living in freshwaters. Consequently, data from predominantly terrestrial tetrapods[4,5] are used to guide environmental policy[6] and conservation prioritization[7], whereas recent proposals for target setting in freshwaters use abiotic factors[8–13]. However, there is evidence[14–17] that such data are insufficient to represent the needs of freshwater species and achieve biodiversity goals[18,19]. Here we present the results of a multi-taxon global freshwater fauna assessment for The IUCN Red List of Threatened Species covering 23,496 decapod crustaceans, fishes and odonates, finding that one-quarter are threatened with extinction. Prevalent threats include pollution, dams and water extraction, agriculture and invasive species, with overharvesting also driving extinctions. We also examined the degree of surrogacy of both threatened tetrapods and freshwater abiotic factors (water stress and nitrogen) for threatened freshwater species. Threatened tetrapods are good surrogates when prioritizing sites to maximize rarity-weighted richness, but poorer when prioritizing based on the most range-restricted species. However, they are much better surrogates than abiotic factors, which perform worse than random. Thus, although global priority regions identified for tetrapod conservation are broadly reflective of those for freshwater faunas, given differences in key threats and habitats, meeting the needs of tetrapods cannot be assumed sufficient to conserve freshwater species at local scales.

Globally, biodiversity is in decline[6] with freshwater ecosystems being particularly affected[20]. On the basis of monitored natural inland wetlands (including peatlands, marshes, swamps, lakes, rivers and pools, among others), 35% of wetland area was lost between 1970 and 2015, at a rate three times faster than that of forests[21]. Of the remaining wetland habitats, 65% are under moderate-to-high levels of threat[22] and 37% of rivers over 1,000 km are no longer free-flowing over their full length[23]. Declines are continuing, generally out of sight and out of mind, despite the importance of the freshwater realm. Freshwaters support over 10% of all known species, including approximately one-third of vertebrates and one-half of fishes, while only covering less than 1% of the surface of the Earth[1]. This diversity of freshwater species provides essential ecosystem services (such as nutrient cycling, flood control and climate change mitigation[2]), can be used as bioindicators of wetland quality[24], and supports the culture, economy and livelihoods of billions of people worldwide[2].

Comprehensive assessments of species extinction risk from the International Union for Conservation of Nature's (IUCN) Red List of Threatened Species (hereafter 'IUCN Red List') are used to document and track trends in the status of biodiversity[25], and inform national-to-global biodiversity strategy, policy and prioritization to halt and reverse species loss[26]. Comprehensive assessments of birds[27], amphibians[28] and mammals[29] have been available for over 20 years, with repeat assessments now available[30–32], and so are the data of choice for global biodiversity science and policy[4–7]. Recently, a global reptile assessment was completed, highlighting the shared conservation needs of all tetrapods[33]. However, generation of global data and assessments for freshwater fishes and invertebrates has received comparatively little investment, political will or attention, including from the mainstream conservation community[34]. This has meant that recent target-setting approaches for freshwater systems have been restricted to the use of abiotic hydrological measures, such as water use and quality[8–13]. Reliance on predominantly terrestrial tetrapod data or freshwater abiotic data in making conservation decisions requires the assumption that these data types serve as effective surrogates for freshwater species. However, evidence has shown that surrogacy of species is generally poorer where the taxonomic group used as a surrogate is from a different environmental realm from that of the target[16,17]. In addition, the efficacy of using abiotic surrogates for freshwater biodiversity remains untested.

Moreover, until recently, the freshwater realm has not been given the same priority as the terrestrial and marine realms in global environmental governance, and has often been included within either terrestrial or marine systems, despite evidence of its distinct management needs (for example, considering connectivity, flow regime and seasonality)[14,15,18,19]. For example, the United Nations Sustainable Development Goals (SDGs) focus principally on terrestrial (SDG 15: life on land) and marine (SDG 14: life below water) biomes, despite the fact that freshwater species are key to achieving these goals[35]. There have been recent advances in highlighting freshwaters as a distinct realm with unique needs and the Kunming-Montreal Global Biodiversity Framework (GBF) specifically calls out inland waters in targets 2 and 3 (ref. 36).

To improve availability of information for use in the conservation and management of freshwater species, we examined the results of the most comprehensive multi-taxon global freshwater fauna assessment to date on the IUCN Red List to summarize the extinction risk, distribution, key habitats and primary drivers of decline of freshwater decapod crustaceans, fishes and odonates (hereafter 'freshwater species'). In addition, to test whether it is appropriate to rely on predominantly terrestrial tetrapod data or freshwater abiotic data when making conservation decisions on freshwater biodiversity at a global scale, we investigated whether threatened tetrapods (amphibians, birds, mammals and reptiles) and two widely used hydrological variables (water stress, representing the ratio of total water demand to available renewable supplies; and water quality, focusing on nitrogen levels, representing eutrophication) are effective surrogates for these threatened freshwater species.

## Assessing extinction risk

We completed a multi-taxon global freshwater fauna assessment using the IUCN Red List categories and criteria[37] to evaluate the extinction risk of 23,496 freshwater species, through completion of global assessment efforts for freshwater fishes and odonates (dragonflies and damselflies), and drawing on previously published IUCN Red List data on freshwater decapod crustaceans (crabs, crayfishes and shrimps)[38–40]. We were unable to include freshwater molluscs in our analysis because only half of the known species globally are currently assessed for the IUCN Red List, with notable geographical biases, meaning their inclusion would introduce regional and taxonomic biases. Where relevant, we highlight the implications of this omission from our dataset below.

Species on the IUCN Red List are placed into categories indicating their extinction risk using a set of five quantitative criteria (A–E), which measure symptoms of risk: (A) population declines; (B) restricted ranges; (C) small and declining populations; (D) very restricted or small populations; and (E) quantitative analysis[37]. Through consultation with species experts (for example, taxonomists, field scientists and fisheries experts), the criteria are applied and validated based on the best-available data at the time of assessment, but with a range of data qualities acceptable to allow broad applicability even to data-poor species. The system is explicitly designed to handle uncertainty. Species experts work with trained IUCN Red List facilitators and follow guidance materials to ensure consistency in application of the criteria between assessments, including across taxonomic groups. The freshwater fauna assessments analysed here were completed over a 20-year period with input from more than 1,000 species experts (Supplementary Note 1) achieved through a combination of over 100 workshops with additional remote assessment and review work (Supplementary Table 1; see Methods for further details of the Red List assessment process and potential associated biases).

The best estimate of the proportion of freshwater species threatened with extinction (considering species assessed as critically endangered, endangered or vulnerable (hereafter 'threatened' species), plus those assessed as extinct in the wild) indicates that close to one-quarter (24%) are at high risk of extinction (Fig. 1 and Extended Data Table 1). This is comparable with tetrapods of which 23% are threatened (Fig. 1). There is variation in extinction risk between the freshwater groups considered, with decapods having the highest percentage of species threatened (30%) compared with 26% for freshwater fishes and 16% for odonates (Fig. 1 and Extended Data Table 1). Our results support previous estimates of proportion of threatened species as calculated through the sampled Red List approach[41] for fishes[42] and odonates[43], through which 25% of 733 assessed freshwater fishes and 14% of 1,500 assessed odonates were reported threatened (following the current methodology for calculating the best estimate of proportion threatened). The sampled Red List for freshwater molluscs[44] found close to one-third threatened with extinction, indicating that the estimates for

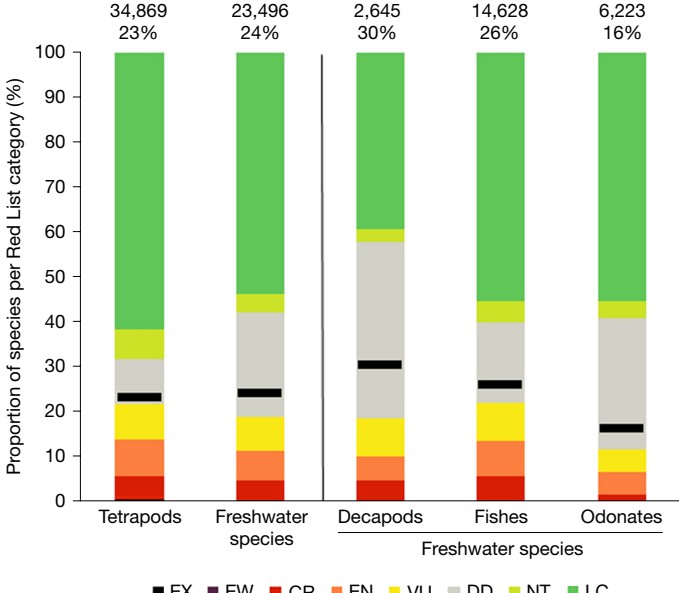

**Fig. 1 | Patterns of extinction risk in tetrapods (combined) and freshwater species (decapod crustaceans, fishes and odonates; combined and individually).** The numbers above each column refer to the total numbers of species assessed and the best estimates of the proportion of species threatened (Methods). The black lines represent the best estimates of the proportion of species threatened. Red List categories are as follows: extinct (EX), extinct in the wild (EW), critically endangered (CR), endangered (EN), vulnerable (VU), data deficient (DD), near threatened (NT) and least concern (LC).

threatened freshwater species presented here are conservative and might increase if freshwater molluscs were included.

Since AD 1500, some 89 (0.4%) assessed freshwater species, including 82 freshwater fishes, 6 decapods and 1 odonate, have been reported to have gone extinct (Extended Data Table 1), with the majority of these native to the USA (22 species, noting this could be a reporting bias, although other data-rich regions (for example, Europe) do not show such high numbers of extinctions), Mexico (15 species) or the Philippines (15 species, all of which were endemic to Lake Lanao, although this could be an artefact of taxonomic uncertainty over species diversity within the lake). Eleven species, all freshwater fishes, have been assessed as extinct in the wild, and survive only as captive populations (Extended Data Table 1), with eight of these originally native to Mexico. The true number of extinctions of freshwater species will probably be higher given the overall lack of research on and long-term monitoring of freshwater biodiversity (see 'Data deficient species' below), as well as the existence of extinct undescribed species that are not assessed on the IUCN Red List. In addition, one-fifth of species assessed as critically endangered (187 out of 949 species) are tagged as either possibly extinct (178 species) or possibly extinct in the wild (9 species; Extended Data Table 1), meaning they are probably already extinct, but there is insufficient evidence to confirm this. These species include 149 freshwater fishes, 19 decapods and 19 odonates, with the highest concentration (46 species) occurring in the Lake Victoria Basin in eastern Africa. The sampled Red List for freshwater molluscs[44] estimated one-quarter of critically endangered species to be possibly extinct.

Close to one-quarter (23%) of freshwater species are assessed as data deficient, indicating that insufficient information is available to assess their extinction risk, compared with only 10% of all tetrapods (Fig. 1 and Extended Data Table 1). Uncertainty introduced by data deficient species (Methods) means that there is much greater variability in the estimate of the proportion of threatened freshwater decapods, fishes and odonates (18–42%) than threatened tetrapods (21–31%; Extended Data Table 1). Within the freshwater groups, there are higher levels of

data deficiency in invertebrates (39% of decapods and 29% of odonates) than in fishes (18%; Fig. 1 and Extended Data Table 1), also leading to greater variability in the estimates of proportion threatened (18–58% of decapods, 11–41% of odonates and 21–40% of fishes; Extended Data Table 1). The sampled Red List for freshwater molluscs[44] estimated data deficiency to fall within our estimates for odonates and decapods (36%).

## Threats

Among freshwater decapods, fishes and odonates, 54% of threatened species are considered to be affected by pollution, 39% by dams and water extraction, 37% by land-use change and associated effects from agriculture (from subsistence to agro-industry scales, excluding aquaculture; note that the threats of pollution and agriculture are strongly linked), and 28% by invasive species and disease (Fig. 2a). Most threatened species (84%) are affected by more than one threat. These threats primarily cause freshwater habitat loss and degradation, and their importance is echoed by the published literature, which also highlights how they act cumulatively[3,45] and are similarly affecting omitted freshwater groups, such as freshwater molluscs[44]. For tetrapods, two key threats dominate, both linked to habitat loss, with agriculture considered to affect 74% of threatened species and logging to affect 49% of threatened species.

Our results support previous evidence that habitat loss is driving extinction risk in odonates[43], with agriculture, logging and urban development affecting 61%, 57% and 29% of threatened odonates, respectively (Fig. 2a). Pollution is also considered a threat, affecting 28% of threatened odonates, primarily in their larval stages. For decapods, pollution is considered the most prevalent threat (60%), suggesting that deteriorating habitat quality is driving extinction risk, with agricultural and forestry effluents (39%, including nutrient loads, herbicides and pesticides, and sedimentation) and domestic and urban waste waters (33%, including run-off and sewage) as the key sources. Pollution is also considered a key threat to fishes (59%), again with agricultural and forestry effluents (45%) as the key sources, followed by domestic and urban waste waters (29%), and industrial and military effluents (21%, including seepage from mining and oil exploration). Water management (including dams and water extraction) is considered another key threat to fishes (46%), with dams blocking migration routes and causing habitat degradation and loss (for example, by affecting downstream hydrology and flow regime, altering sediment flux and converting upstream riverine habitat to lentic impoundments)[46]. Fishes face the widest variety of threats, also being frequently affected by invasive species and disease (33%), agriculture (32%) and overfishing (27%), with the latter predominantly linked to targeted harvesting.

We found that the most frequent causes of extinction in freshwater species do not directly reflect those outlined above, highlighting the increased severity of effects of some threats, although patterns differ depending on whether species flagged as possibly extinct and possibly extinct in the wild are considered extant or extinct. In both scenarios, invasive species and disease, and overharvesting have contributed to more extinctions than would be expected based on the proportion of threatened species affected by them (Extended Data Table 2). It should be noted that most extinctions are thought to be caused by multiple and often interacting stressors, with over two-thirds of extinct species having more than one threat recorded. Looking pairwise at threats, dams and water management in combination with pollution or invasive species and disease were recorded as the most common joint drivers of extinction. These threats each paired with overharvesting were also recorded as frequent causes of extinctions (Extended Data Table 3).

Close to one-fifth of threatened freshwater species are recorded to be affected by climate change and severe weather events (Fig. 2a). Climate change negatively affects freshwater ecosystems both directly, for example, by shifting temperatures, flow regimes and leading to more severe weather events[47], and indirectly, for example, by amplifying other threats, notably invasions and increased human demand for water storage. Globally, this is an intensifying threat[30,45,47] and we expect its proportional effect to increase over time. Owing to a lack of modelling studies on the effect of climate change on freshwater species, its effect may be underestimated in current IUCN Red List assessments, and new approaches in evaluating the extinction risk posed by climate change may be required in future assessment efforts[48].

## Habitats

Unsurprisingly, natural inland wetlands are found to be the key habitats for freshwater species. Forests are also considered a key habitat for odonates (particularly (sub)tropical species), providing shelter and hunting grounds for the adult life stages of 74% of threatened odonates, whereas the larval stages are generally found in wetland habitats, both embedded in and found outside of forests. Forests and inland wetlands are also the most important habitats for tetrapods, although with the order of importance reversed, supporting 82% and 28% of threatened species, respectively (Extended Data Fig. 1).

In terms of natural inland wetland habitats, permanent rivers are considered the most commonly used, supporting 71% of threatened freshwater species (Fig. 2b). Freshwater fishes rely on the widest variety of wetland habitats, including permanent rivers (hosting 70% of threatened species), permanent lakes (23%), seasonal rivers (13%) and permanent pools (11%), noting that many species occur in multiple habitats. Karst hosts consistently more threatened species than would be expected based on total species. This is particularly true for freshwater decapods; 24% of threatened decapods occur in this habitat (Fig. 2b), but only 12% of all decapods (chi-squared test, $P < 0.001$), probably because of the richness of small, restricted range populations of decapods in karst, in combination with various threats to the habitat (for example, degradation due to exploitation for resources, from recreation or pollution). More extinctions of freshwater species have been reported from permanent lakes and from springs and oases than would be expected based on use of these habitats by threatened species, with species endemic to these habitats (the latter often also highly restricted) having no way to escape from prominent threats, such as invasive species, water extraction and harvesting. This result holds whether species flagged as possibly extinct and possibly extinct in the wild are considered extant or extinct (Extended Data Table 4).

## Spatial patterns

Richness of freshwater species is highest in the tropics, with concentrations in the Amazon basin in South America; western, central and eastern Africa; and tropical Asia from southern India and Sri Lanka through Sundaland to coastal New Guinea (Extended Data Fig. 2a). Outside of the tropics, similarly high levels of freshwater species richness are also found in the eastern USA (Extended Data Fig. 2a). Richness of data deficient species (Extended Data Fig. 2b) generally reflects that of overall species richness.

Concentrations of threatened species richness are smaller and are patchily distributed, but with Lake Victoria (Kenya, Tanzania and Uganda), Lake Titicaca (Bolivia and Peru), Sri Lanka's Wet Zone and Western Ghats (India) having the greatest absolute richness of threatened freshwater species (Fig. 3). Lake Titicaca, Chile's Biobío region and the Azores (Portugal) all have very high threatened species richness relative to absolute species richness (Extended Data Fig. 3a). These spatial patterns hold whether possibly extinct and possibly extinct in the wild species are considered extant or extinct (Extended Data Figs. 2c and 3b).

## Surrogates for freshwater species

We investigated the degree to which threatened tetrapods and freshwater abiotic factors serve as effective surrogates for threatened

**a**

| Threat | Threatened freshwater species | Extinct freshwater species | Threatened decapods | Threatened fishes | Threatened odonates | Threatened tetrapods |
|---|---|---|---|---|---|---|
| Pollution | 0.54 | 0.52 | 0.60 | 0.59 | 0.28 | 0.17 |
| Dams and water management | 0.39 | 0.63 | 0.19 | 0.46 | 0.23 | 0.09 |
| Agriculture | 0.37 | 0.06 | 0.33 | 0.32 | 0.61 | 0.74 |
| Invasive species and disease | 0.28 | 0.55 | 0.24 | 0.33 | 0.07 | 0.33 |
| Logging | 0.25 | 0.04 | 0.19 | 0.19 | 0.57 | 0.49 |
| Urban development | 0.23 | 0.07 | 0.34 | 0.19 | 0.29 | 0.35 |
| Hunting and fishing | 0.21 | 0.37 | 0.15 | 0.27 | 0.02 | 0.29 |
| Energy production and mining | 0.18 | 0.01 | 0.10 | 0.20 | 0.16 | 0.18 |
| Climate change and severe weather | 0.18 | 0.05 | 0.14 | 0.19 | 0.16 | 0.23 |
| Human intrusions and disturbance | 0.08 | 0.02 | 0.18 | 0.06 | 0.08 | 0.11 |
| Other ecosystem modifications | 0.06 | 0.04 | 0.03 | 0.08 | 0.02 | 0.02 |
| Transportation | 0.05 | 0.01 | 0.07 | 0.06 | 0.01 | 0.13 |
| Fire and fire suppression | 0.05 | 0.00 | 0.08 | 0.03 | 0.12 | 0.19 |
| Problematic native species | 0.04 | 0.02 | 0.02 | 0.06 | 0.01 | 0.06 |
| Aquaculture | 0.02 | 0.00 | 0.00 | 0.02 | 0.06 | 0.01 |
| Geological events | 0.01 | 0.00 | 0.00 | 0.01 | 0.01 | 0.02 |

**b**

| Habitat | Threatened freshwater species | Extinct freshwater species | Threatened decapods | Threatened fishes | Threatened odonates |
|---|---|---|---|---|---|
| Permanent rivers | 0.71 | 0.37 | 0.65 | 0.70 | 0.77 |
| Permanent lakes | 0.18 | 0.48 | 0.10 | 0.23 | 0.02 |
| Seasonal rivers | 0.10 | 0.00 | 0.02 | 0.13 | 0.03 |
| Permanent pools | 0.09 | 0.03 | 0.04 | 0.11 | 0.07 |
| Bogs and marshes, among others | 0.08 | 0.04 | 0.07 | 0.08 | 0.08 |
| Seasonal pools | 0.08 | 0.00 | 0.02 | 0.10 | 0.02 |
| Springs and oases | 0.06 | 0.19 | 0.03 | 0.07 | 0.03 |
| Karst | 0.05 | 0.00 | 0.24 | 0.03 | 0.00 |
| Seasonal lakes | 0.02 | 0.00 | 0.01 | 0.02 | 0.01 |
| Other wetlands | 0.01 | 0.00 | 0.01 | 0.02 | 0.00 |
| Saline, brackish or alkaline | 0.01 | 0.00 | 0.00 | 0.02 | 0.00 |

**Fig. 2 | Proportion of threatened freshwater species, extinct freshwater species and threatened tetrapods. a**, Proportion of threatened freshwater species (decapod crustaceans, fishes and odonates; combined and individually), extinct freshwater species (combined) and threatened tetrapods (combined) affected by each threat. The darker cells indicate a greater proportion of species affected by the threat. Threats are not mutually exclusive. Threats are coded following the IUCN Threats Classification Scheme (version 3.3) and combined for presentation as follows (the value of the highest hierarchical level is indicated; all subsequent levels are included): pollution (9); dams and water management (7.2); agriculture (2.1, 2.2 and 2.3); invasive species and disease (8.1, 8.3, 8.4, 8.5 and 8.6); logging (5.2 and 5.3); urban development (1); hunting and fishing (5.1 and 5.4); energy production and mining (3); climate change and severe weather (11); human intrusions and disturbance (6); other ecosystem modifications (7.3); transportation (4); fire and fire suppression (7.1); problematic native species (8.2); aquaculture (2.4); and geological events (10). For the number of species: threatened freshwater species $n = 4,190$, extinct freshwater species $n = 82$, threatened decapods $n = 472$, threatened fishes $n = 3,032$, threatened odonates $n = 686$ and threatened tetrapods $n = 7,112$. **b**, Proportion of threatened freshwater species (decapod crustaceans, fishes and odonates; combined and individually) and extinct freshwater species (combined) using each wetland habitat. The darker cells indicate a greater proportion of species using the habitat. Habitats are not mutually exclusive. Habitats are coded following the IUCN Habitats Classification Scheme (version 3.1) as follows: permanent rivers (5.1); permanent lakes (5.5); seasonal rivers (5.2); permanent pools (5.7); bogs and marshes, among others (5.4); seasonal pools (5.8); springs and oases (5.9); karst (5.18); seasonal lakes (5.6); other wetlands (5.3, 5.10, 5.11 and 5.12); and saline, brackish or alkaline (5.14, 5.15, 5.16 and 5.17). For the number of species: threatened freshwater species $n = 4,255$, extinct freshwater species $n = 100$, threatened decapods $n = 484$, threatened fishes $n = 3,071$ and threatened odonates $n = 700$. In panels **a**,**b**, threatened species include those assessed as critically endangered (including those flagged as possibly extinct and possibly extinct in the wild), endangered or vulnerable. Extinct freshwater species include those assessed as extinct or extinct in the wild.

freshwater species targets in spatial conservation planning. Using a complementarity representation approach (that is, ensuring areas selected complement those already chosen), we applied two strategies, which together highlight key areas of conservation priority for threatened species: strategy a to prioritize rarity-weighted threatened species richness (that is, to prioritize inclusion of areas containing many threatened species with restricted ranges), and strategy b to maximize inclusion of the most range-restricted species (that is, to prioritize inclusion of areas core to the most range-restricted species; see Methods for more details). We derived species accumulation indices (SAIs) to test surrogate effectiveness, with values approaching one indicating strong surrogacy, zero values indicating random surrogacy, and negative values indicating surrogacy worse than random. We used the following descriptors to define SAI performance: 0.01–0.19 as very poor, 0.20–0.39 as poor, 0.40–0.59 as reasonable, 0.60–0.79 as good, and 0.80–0.99 as very good. For reference, surrogacy across 464 tests from 16 studies[17] yielded median SAI = 0.12, and 8 tests from a single study using terrestrial surrogates and freshwater targets yielded median SAI = 0.38. The SAI approach is more appropriate in addressing the extent to which areas selected for surrogates capture targets than approaches based on spatial congruence.

At the scale investigated (approximately 50 × 50 km resolution), when prioritizing rarity-weighted threatened species richness, we found tetrapods (combined) to be good surrogates for freshwater species (combined) as targets (SAI = 0.66), although individually tetrapod classes are only reasonable surrogates (amphibians SAI = 0.45, birds SAI = 0.51, mammals SAI = 0.55 and reptiles SAI = 0.56; Extended Data Fig. 4a). Tetrapods (combined) are good surrogates for each individual freshwater group, with efficacy increasing from fishes (SAI = 0.60) to decapods (SAI = 0.73) to odonates (SAI = 0.80; Extended Data Fig. 5a), presumably explained by the reliance of many threatened decapods and odonates on forest habitat, similar to many threatened tetrapods (Extended Data Fig. 1).

When maximizing inclusion of the most range-restricted species, threatened tetrapods (combined) are weaker, although still reasonable, surrogates for threatened freshwater species (combined; SAI = 0.58; Extended Data Fig. 4b). Again, individual tetrapod classes perform less well as surrogates than tetrapods (combined), with mammals serving as reasonable surrogates (SAI = 0.42), followed by reptiles (SAI = 0.37), birds (SAI = 0.35) and amphibians as the poorest surrogates (SAI = 0.21; Extended Data Fig. 4b). This suggests that threatened freshwater species with the smallest ranges tend to be in locations that differ from those of tetrapods with the smallest ranges, particularly so for amphibians, which have the narrowest ranges among tetrapods. When looking at freshwater groups individually as targets with tetrapods (combined) as surrogates, surrogacy again increases from fishes (SAI = 0.49) to decapods (SAI = 0.67) to odonates (SAI = 0.81; Extended Data Fig. 5b).

All SAI values for the abiotic factors, representing water quantity and quality, are negative, meaning performance is worse than expected from random solutions (Extended Data Fig. 6). The effectiveness is equally bad for both conservation strategies. Water stress and eutrophication are, therefore, very poor surrogates for threatened freshwater species. This suggests that relying on these surrogates for conservation and management decisions could lead to suboptimal or even harmful outcomes and, therefore, conservation strategies that rely on abiotic indicators should be re-evaluated.

## Discussion

This study marks the completion of a systematic assessment of the global extinction risk of multiple freshwater fauna groups, through a large-scale expert consultation process following quantitative criteria, and provides a vital step in addressing the decline in freshwater biodiversity globally. Given that around one-quarter of freshwater species are at high risk of extinction, with 89 confirmed and an additional 178 suspected extinctions since 1500, there is urgency to act quickly to address threats to prevent further species declines and losses. Lack of data on the status and distribution of freshwater biodiversity can no longer be used as an excuse for inaction. Although agriculture and invasive species are considered to be major threats to both freshwater species and tetrapods, some threats are found to be more prevalent to freshwater species, notably pollution, dams and water extraction, and as such they require targeted actions in response. These primary threats are systemic in their effects and will require changes in water management practices at a catchment scale, over and above species-specific or site-based actions. Integration with the water sector and improved consideration of biodiversity in water development and governance, for example, through ecological stakeholder analogue or healthy watersheds approaches, is, therefore, an essential component of tackling declines in freshwater species. This connection and collaboration should be solutions focused, for example, using nature-based solutions to offer developments that simultaneously benefit biodiversity and human well-being[34]. Important consideration should additionally be given to tackling overharvesting given that this is considered a particularly prominent threat in driving freshwater species extinctions.

The majority of threatened freshwater species are considered at risk because of continuing declines or plausible threats within their restricted distributions (that is, 90% are assessed based on IUCN Red List criteria B or D2 (ref. 37)), rather than due to small population size, decline or quantitative analysis (that is, IUCN Red List criteria A, C, D1 and E[37]). This is in part because freshwater habitats are often fragmented, precluding many species travelling between isolated habitats and resulting in many naturally range-restricted species, but also in part because population data are missing for most freshwater species. Population data would assist in assigning species to categories other than data deficient, thereby reducing overall uncertainty regarding extinction risk. This strongly underscores the need for increased investment into quantitative research and monitoring of freshwater species[49] to decrease uncertainty around overall extinction risk, noting that pressures from human populations are likely to increase with time as existing threats are exacerbated and new threats emerge[45]. In addition, monitoring will increase the evidence base for the efficacy (or not) of particular actions for tackling threats and improving the status of species to better guide future action[34,50,51]. Increased involvement of stakeholders beyond conservation scientists (for example, natural resource managers, infrastructure developers and local communities) and improved regulation will greatly increase the volume of data, as well as the relevancy and legitimacy of the data for all stakeholders[34,52]. Citizen science schemes could be a potential solution given the high need for data, but low availability of financial resources to support their generation[53]. Newer survey techniques should be investigated, such as use of environmental DNA[54], in addition to conventional sampling. Data generated by increased monitoring efforts are vital to lower the proportion of data deficient species, including by providing information on overlooked taxa that could be at high risk of extinction owing in part to their isolation (Extended Data Fig. 2b).

Contrary to previously published evidence[16,17], our analysis found that globally threatened tetrapods act as good surrogates for threatened freshwater species, and as reasonable surrogates for range-restricted threatened freshwater species, noting that over one-quarter of threatened tetrapods are directly dependent on natural wetland habitats. This suggests that broad-scale conservation priority regions (in line with the resolution investigated here) based on terrestrial species groups effectively represent threatened freshwater species overall, although many highly range-restricted freshwater species, especially among fishes, are unlikely to be prioritized incidentally through co-occurrence with tetrapods. Ancient watersheds, springs and karst systems, for example, often host large numbers of threatened endemic species from freshwater groups (including freshwater molluscs, which were omitted here[44]), but few tetrapods, and thus require specific targeted action. In addition, distribution of species within their ranges will not

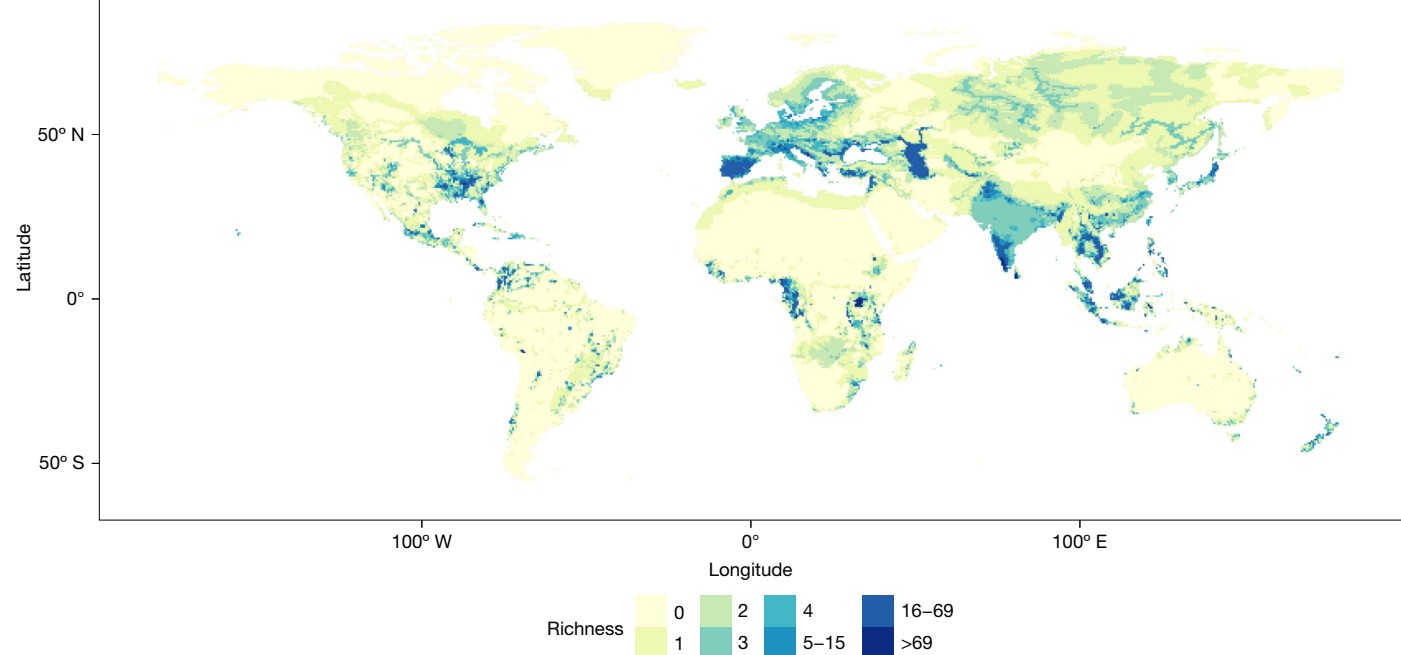

**Fig. 3 | Absolute richness of threatened freshwater species.** Threatened species are those assessed as critically endangered, endangered or vulnerable (including those flagged as possibly extinct and possibly extinct in the wild). The following distributions are included: presence refers to extant, probably extant or possibly extinct; origin denotes native, reintroduced or assisted colonization; and seasonality indicates resident, breeding, non-breeding or passage. The value of each cell is calculated as the count of threatened species with a mapped distribution overlapping the cell. Richness is shown using a 0.5 × 0.5 latitude–longitude grid and WGS84. World Bank Official Boundaries (licensed under a Creative Commons licence CC BY 4.0) were used as the base map. For absolute threatened species richness of tetrapods, see ref. 33.

be uniform, and the disproportionate importance of some areas and/or habitat types, for example, those used as feeding or breeding grounds, needs to be considered when planning action on the ground.

Where there are areas of high spatial overlap between tetrapod and freshwater species conservation priorities, it is important that both groups are actively included in management and conservation action plans. Our analysis has highlighted differences in key habitats and threats between the two groups, such that meeting the needs of tetrapods cannot be assumed as sufficient to conserve freshwater species at local scales. In addition, action plans need to consider the effects of hydrological connectivity, environmental flows and habitat structure, because without these explicit considerations, most actions are less effective for freshwater species[55]. Encouragingly, conservation initiatives that integrate terrestrial and freshwater biodiversity in planning and prioritization show substantial benefits to freshwater species, with negligible reduction in benefits to terrestrial species[56].

Our results reveal that water stress and eutrophication are very poor surrogates when used in conservation planning for threatened freshwater species. The distribution of biodiversity is influenced by a complex interplay between various biotic and abiotic factors, and conservation strategies that rely solely on abiotic factors as key indicators should be re-evaluated. This analysis marks, to our knowledge, the first global investigation into assessing abiotic environmental data as surrogates for freshwater species, and is consistent with previous findings[17] for terrestrial species that environmental surrogates are much poorer than cross-taxonomic surrogates. As the private and public sectors rush to set science-based targets such as those in the GBF[36] to help to deliver environmental goals, our results suggest that setting targets around non-living nature will not be sufficient to protect and conserve living nature and may be harmful in terms of its opportunity cost and displacement of threats to more important places for freshwater biodiversity.

As the next step, it is important that availability of this new dataset is communicated to all relevant stakeholders, ranging from local conservation practitioners, to the public sector and national bodies in planning developments, to global policy instruments. In addition, these stakeholders should be supported to maximize uptake and integration of the dataset into their activities, to connect science to evidence-based management and conservation actions benefitting freshwater species[34]. For example, private sector users of the Integrated Biodiversity Assessment Tool (IBAT) can run a freshwater report that uses IUCN Red List data to provide information on freshwater species (highlighting threatened and migratory species) found in watersheds upstream and downstream of points of interest, to help to mitigate risk to freshwater biodiversity when planning developments. IBAT also incorporates IUCN Red List data on freshwater species into country profiles, which can assist countries in reporting on their national biodiversity strategies and action plans. The integration of freshwater biodiversity data should be cross-sectorial (involving coordination with agriculture and energy sectors, for example, given their direct use of and effects on freshwater systems), as healthy freshwater ecosystems maintain ecosystem services and support communities and human livelihoods worldwide. Conservation of freshwater fish species in particular is vital in regions where communities depend on them for their protein needs; otherwise, food security, and related livelihoods and economies, will be compromised[2]. Although some knowledge systems, notably those of Indigenous communities, already recognize the inherent connections between humans and freshwaters[57], there is a need to reshape the relationship of society as a whole to move understanding away from freshwaters solely being a resource for exploitation and a component of the terrestrial landscape[34].

The new global freshwater IUCN Red List dataset that we introduced here, combined with upcoming advances in calculating area of habitat for freshwater species (F. A. Ridley et al., unpublished data), will soon enable integration of the assessed freshwater groups into the species threat abatement and restoration (STAR) metric[4]. This will allow evaluation of the contribution of particular actions at specific locations to reducing species extinction risk[4], enabling establishment of science-based targets for freshwater species conservation, aligned with GBF goal A[36] and with SDG 15. In support of the GBF targets, the new

global freshwater IUCN Red List dataset can also contribute as a starting point for the identification of Key Biodiversity Areas, sites important for the global persistence of biodiversity[58] for freshwater species. These have already been identified in some regions where comprehensive data are available (for example, see ref. 59). Moreover, the data can help to inform countries whose freshwater biodiversity status makes them most critical for restoration, as part of their commitments to the Freshwater Challenge, a country-driven initiative aimed at leveraging the support needed to bring 300,000 km of rivers and 350 million hectares of inland waters under restoration by 2030. In addition, the data presented here will allow better determination of where existing protected areas, or other protection and management mechanisms, are in areas of high freshwater biodiversity importance and to identify and prioritize gaps for further action. This has been done for Africa[16], but now can be repeated globally. The dataset will also form the basis of (multi-)species conservation planning processes across the globe.

Although the freshwater groups analysed here provide a greatly improved picture of the global status and distribution of freshwater biodiversity, it is essential that extinction risk assessments of freshwater species expand taxonomic coverage to more fully represent the biodiversity of the realm. This includes a number of ongoing and emerging initiatives to improve our knowledge of: freshwater molluscs, a group that existing research indicates is likely to be highly threatened[44]; caddisflies, stoneflies and mayflies, which are typically used in biomonitoring to infer changes in water quality and ecosystem health[24]; and wetland-dependent plants and freshwater fungi, to expand coverage to other kingdoms. In addition, the IUCN Red List Index[25] is a widely used indicator of the status of biodiversity[6], but it relies on taxonomic groups (or samples of them[41]) being assessed more than once to assess trends. Timely reassessments of the freshwater groups are, therefore, essential to track their changing status and to keep in line with new discoveries and taxonomic changes. In addition, GBF goal A[36] tasks countries with using Red List indices as an indicator of national-level progress. Countries should be encouraged and equipped to carry freshwater Red Listing forwards through regular national-level reassessments to meet reporting targets and inform their decision making. Reassessments, however, hold their own challenges due to the lack of freshwater-monitoring capacity in many parts of the world[49], limiting our ability to provide new data to update assessments. Expanding this capacity is, therefore, vital to track the changing status of freshwater biodiversity.

Complementary to the IUCN Red List, assessments using the recent Green Status framework on species' progress towards recovery will aid in incentivizing conservation action[51]. At present, only two species (giant river prawn, *Macrobrachium rosenbergii*, and Acıgöl killifish, *Anatolichthys transgrediens*) in the freshwater groups considered here have published Green Status assessments, although the IUCN Red List provides evidence that the extinction risk status of a number of freshwater species has genuinely improved as a result of effective, targeted conservation efforts. Finally, there is also a need to expand global assessment efforts to the level of wetland ecosystems, through the IUCN Red List of Ecosystems[60]. Therefore, despite the progress demonstrated here, increased investment is clearly required into currently under-funded[34] research on and IUCN Red Listing of freshwater biodiversity, supporting evidence-based management and conservation actions, monitoring the effects of these actions, and processes to feed these results back into assessment efforts.

It is encouraging that recent global environmental frameworks have specifically identified a need to act on and restore freshwater species and habitats to meet commitments for reversing global declines in biodiversity. This is a key step in restoring freshwater biodiversity[34]. Moving forwards, we hope this global freshwater IUCN Red List dataset, as well as future expanded iterations, and recommendations from this analysis will be integrated into evidence-based conservation and management actions and policy measures from local to global scales to help to bend the curve for freshwater biodiversity loss[15].

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

Catherine A. Sayer[1]✉, Eresha Fernando[1], Randall R. Jimenez[2], Nicholas B. W. Macfarlane[3], Giovanni Rapacciuolo[4], Monika Böhm[5], Thomas M. Brooks[6], Topiltzin Contreras-MacBeath[7], Neil A. Cox[3,8], Ian Harrison[9], Michael Hoffmann[10], Richard Jenkins[1], Kevin G. Smith[1], Jean-Christophe Vié[11], John C. Abbott[12], David J. Allen[1], Gerald R. Allen[13], Violeta Barrios[14], Jean-Pierre Boudot[15], Savrina F. Carrizo[16], Patricia Charvet[17], Viola Clausnitzer[18], Leonardo Congiu[19], Keith A. Crandall[20], Neil Cumberlidge[21], Annabelle Cuttelod[22], James Dalton[6], Adam G. Daniels[23], Sammy De Grave[24], Geert De Knijf[25], Klaas-Douwe B. Dijkstra[26], Rory A. Dow[26,27], Jörg Freyhof[28], Nieves García[29], Joern Gessner[30], Abebe Getahun[31], Claudine Gibson[32], Matthew J. Gollock[10], Michael I. Grant[33,34], Alice E. R. Groom[35], Michael P. Hammer[36], Geoffrey A. Hammerson[37], Craig Hilton-Taylor[1], Laurel Hodgkinson[38], Robert A. Holland[39], Rima W. Jabado[33,40], Diego Juffe Bignoli[41], Vincent J. Kalkman[26], Bakhtiyor K. Karimov[42], Jens Kipping[43], Maurice Kottelat[44], Philippe A. Lalèyè[45], Helen K. Larson[36], Mark Lintermans[46,47], Federico Lozano[48], Arne Ludwig[49], Timothy J. Lyons[50], Laura Máiz-Tomé[1], Sanjay Molur[52], Heok Hee Ng[53], Catherine Numa[54], Amy F. Palmer-Newton[55], Charlotte Pike[10], Helen E. Pippard[56], Carla N. M. Polaz[57], Caroline M. Pollock[1], Rajeev Raghavan[58], Peter S. Rand[59], Tsilavina Ravelomanana[60], Roberto E. Reis[61], Cassandra L. Rigby[33], Janet A. Scott[1], Paul H. Skelton[62], Matthew R. Sloat[63], Jos Snoeks[64], Melanie L. J. Stiassny[65], Yoshinori Taniguchi[66], Eva B. Thorstad[67], Marcelo F. Tognelli[68], Armi G. Torres[69], Yan Torres[17], Denis Tweddle[62], Katsutoshi Watanabe[70], James R. S. Westrip[1], Emma G. E. Wright[71], E Zhang[72] & William R. T. Darwall[73]

[1]IUCN (International Union for Conservation of Nature), Cambridge, UK. [2]IUCN (International Union for Conservation of Nature), San Jose, Costa Rica. [3]IUCN (International Union for Conservation of Nature), Washington, DC, USA. [4]Elimia, San Diego, CA, USA. [5]Global Center for Species Survival, Indianapolis Zoo, Indianapolis, IN, USA. [6]IUCN (International Union for Conservation of Nature), Gland, Switzerland. [7]Laboratorio de Ictiología, Centro de Investigaciones Biológicas, Universidad Autónoma del Estado de Morelos, Cuernavaca, México. [8]Conservation International, Washington, DC, USA. [9]Free Flowing Rivers Laboratory, Northern Arizona University, Flagstaff, AZ, USA. [10]Zoological Society of London, London, UK. [11]Fondation Franklinia, Geneva, Switzerland. [12]The University of Alabama, Tuscaloosa, AL, USA. [13]Western Australian Museum, Perth, Western Australia, Australia. [14]Sahara Conservation, Saint Maur des Fossés, France. [15]University of Nancy/CNRS, Vandoeuvre-lès-Nancy, France. [16]Zoo and Aquarium Association Australasia, Sydney, New South Wales, Australia. [17]Federal University of Ceará, Fortaleza, Brazil. [18]Senckenberg, Görlitz, Germany. [19]University of Padova, Padova, Italy. [20]George Washington University, Washington, DC, USA. [21]Northern Michigan University, Marquette, MI, USA. [22]Independent researcher, Pully, Switzerland. [23]Independent researcher, Cambridge, UK. [24]Oxford University Museum of Natural History, Oxford, UK. [25]Research Institute for Nature and Forest (INBO), Brussels, Belgium. [26]Naturalis Biodiversity Center, Leiden, The Netherlands. [27]Institute of Biodiversity and Environmental Conservation, Universiti Malaysia Sarawak, Sarawak, Malaysia. [28]Museum für Naturkunde, Leibniz Institute for Evolution and Biodiversity Science, Berlin, Germany. [29]Independent researcher, Malaga, Spain. [30]Leibniz Institute for Freshwater Ecology and Inland Fisheries, Berlin, Germany. [31]Addis Ababa University, Addis Ababa, Ethiopia. [32]Auckland Zoo, Auckland, New Zealand. [33]Centre for Sustainable Tropical Fisheries and Aquaculture and College of Science and Engineering, James Cook University, Townsville, Queensland, Australia. [34]Faculty of Marine Science and Fisheries, Hasanuddin University, Makassar, Indonesia. [35]Royal Society for the Protection of Birds, Sandy, UK. [36]Museum and Art Gallery of the Northern Territory, Darwin, Northern Territory, Australia. [37]NatureServe, Port Townsend, WA, USA. [38]Animals Asia Foundation, Bedford, UK. [39]University of Southampton, Southampton, UK. [40]Elasmo Project, Dubai, United Arab Emirates. [41]Durrell Institute for Conservation and Ecology (DICE), University of Kent, Canterbury, UK. [42]Tashkent Institute of Irrigation and Agricultural Mechanization Engineers National Research University (TIIAME NRU), Tashkent, Uzbekistan. [43]BioCart Ökologische Gutachten, Taucha/Leipzig, Germany. [44]Independent researcher, Delémont, Switzerland. [45]University of Abomey-Calavi, Cotonou, Benin. [46]Centre for Applied Water Science, University of Canberra, Canberra, Australian Capital Territory, Australia. [47]Fish Fondler Pty Ltd, Bungendore, New South Wales, Australia. [48]Laboratorio de Biodiversidad y Genética Ambiental — UNDAV, Avellaneda, Argentina. [49]Leibniz Institute for Zoo and Wildlife Research, Department of Evolutionary Genetics & Humboldt University Berlin, Faculty of Life Sciences, Thaer-Institute for Agricultural and Horticultural Sciences, Berlin, Germany. [50]Center for Species Survival: New Mexico, New Mexico BioPark Society, Albuquerque, NM, USA. [51]Mott MacDonald Environment and Social Division (ENS), Cambridge, UK. [52]Zoo Outreach Organisation, Coimbatore, India. [53]Lee Kong Chian Natural History Museum, National University of Singapore, Singapore, Singapore. [54]IUCN (International Union for Conservation of Nature), Málaga, Spain. [55]Independent researcher, London, UK. [56]Independent researcher, Suva, Fiji. [57]Chico Mendes Institute (ICMBio), Pirassununga, Brazil. [58]Kerala University of Fisheries and Ocean Studies (KUFOS), Kochi, India. [59]Prince William Sound Science Center, Cordova, AK, USA. [60]Mention Zoologie et Biodiversité Animale, Université d'Antananarivo, Antananarivo, Madagascar. [61]Pontifícia Universidade Católica do Rio Grande do Sul, Porto Alegre, Brazil. [62]South African Institute for Aquatic Biodiversity, Makhanda, South Africa. [63]Wild Salmon Center, Portland, OR, USA. [64]Royal Museum for Central Africa, Tervuren and KU Leuven (Leuven University), Leuven, Belgium. [65]American Museum of Natural History, New York, NY, USA. [66]Meijo University, Nagoya, Japan. [67]Norwegian Institute for Nature Research, Trondheim, Norway. [68]American Bird Conservancy, The Plains, VA, USA. [69]Department of Biological Sciences, College of Science and Mathematics, Mindanao State University–Iligan Institute of Technology, Iligan City, Philippines. [70]Division of Biological Sciences, Graduate School of Science, Kyoto University, Kyoto, Japan. [71]Joint Nature Conservation Committee, Peterborough, UK. [72]Institute of Hydrobiology, Chinese Academy of Sciences, Wuhan, China. [73]Tamar Valley National Landscape, Gunnislake, UK. ✉e-mail: catherine.sayer@iucn.org

## Methods

### Data compilation

We used the IUCN Red List categories and criteria[37,61], together with methods developed in other global assessment efforts[28,31,33], to assess the extinction risk of freshwater species. The detailed processes for the assessment of freshwater fishes and odonates are described below. The global assessment efforts for freshwater decapod crustaceans (focused on primary (that is, those that spend their entire life cycle in freshwaters) freshwater crab, crayfish and shrimp species) have been reported previously[38–40]. However, some freshwater decapod species have been assessed or reassessed since these global assessment efforts were first completed. Therefore, updated Red List assessment figures are provided below to reflect the version of the dataset used in this analysis.

For all taxonomic groups considered in this analysis, we used the 2022-2 (December 2022) version[62] of the tabular and spatial data from the IUCN Red List website, which was downloaded in March 2023 (see the 'Red List assessment process' and 'Distribution maps' sections below for a description of the data). This included data on 13,259 freshwater fish species in the classes Actinopterygii, Cephalaspidomorphi, Chondrichthyes, Myxini and Sarcopterygii. Freshwater fish species are defined here as those that spend all or a critical part of their life cycle in freshwaters, and are identified with the 'Freshwater =Inland waters' systems code on the IUCN Red List. In addition, we included preliminary (defined below) IUCN Red List categories for 1,577 freshwater fish species, including 208 reassessments. Together, this covered 14,628 freshwater fish species, representing 79% of the 18,432 formally described freshwater fish species as of December 2022 (ref. 63). The IUCN Red List dataset also included other freshwater-dependent species: 6,223 odonate (dragonfly and damselfly) species (order Odonata), representing 97% of the 6,392 formally described species as of December 2022 (ref. 64); and 2,645 decapod crustacean (primary freshwater crab, crayfish and shrimp) species (families Alpheidae, Astacidae, Atyidae, Cambaridae, Desmocarididae, Euryrhynchidae, Gecarcinucidae, Palaemonidae, Parastacidae, Potamidae, Potamonautidae, Pseudothelphusidae, Trichodactylidae, Typhlocarididae and Xiphocarididae), representing 81% of the 3,267 formally described species as of December 2022 (ref. 65). Together, the assessed species represent 84% of the formally described species in all three taxonomic groups (see the 'Missing species' section below for a description of the gaps in coverage). These groups were selected for assessment because they cover both vertebrates and invertebrates, illustrate a range of life histories, occupy various wetland habitats, are not restricted to particular continents and are groups for which there was thought to be a reasonable level of existing information, such that their comprehensive assessment would provide a good indication of the status of freshwater fauna globally.

This freshwater species dataset was compared with that of the tetrapods: amphibians (class Amphibia; 7,468 species), birds (class Aves; 11,188 species), mammals (class Mammalia; 5,973 species) and reptiles (class Reptilia; 10,222 species).

### Red List assessment process

We worked with species experts and members of the IUCN Species Survival Commission (SSC) to compile data, draft and review Red List assessments.

For the freshwater fishes, the Red List assessments were primarily completed through a series of IUCN-led regional sub-projects, through which global IUCN Red List assessments of all species native to the region were completed. There were 40 regional sub-projects between 2003 and 2023, which included close to 60 workshops (Supplementary Table 1a). Some families (that is, Acipenseridae, Anguillidae and Salmonidae) were assessed primarily through separate processes facilitated by the relevant IUCN SSC Specialist Group (that is, sturgeon, anguillid eel and salmon, respectively). Overall, the freshwater fish Red List assessments were drafted by 820 species experts (Supplementary Note 1a) with additional individuals contributing through facilitation and review.

For the odonates, the Red List assessments were also primarily completed through a series of regional sub-projects (led by either IUCN, members of the IUCN SSC Dragonfly Specialist Group or related groups, for example, the Sociedad de Odonatología Latinoamericana), through which global IUCN Red List assessments of all species native to the region were completed. There were 26 regional sub-projects between 2003 and 2023, which included close to 50 workshops (Supplementary Table 1b). In addition, between 2006 and 2008, the sampled approach to the Red List Index project[43] randomly selected and assessed 1,500 odonate species. Finally, work through the IUCN-Toyota Partnership[66] filled remaining gaps (primarily in Latin America and Southeast Asia) to complete the global dragonfly assessment. Overall, the odonate Red List assessments were drafted by 112 species experts (Supplementary Note 1b) with additional individuals contributing through facilitation and review.

Global assessment efforts for freshwater decapod crustaceans have been previously reported[38–40]. Overall, these assessments were drafted by 78 species experts (Supplementary Note 1c) with additional individuals contributing through facilitation and review.

For all freshwater species, species experts and/or project staff first sourced information for each species from the published and grey literature, online databases, and expert knowledge and collections, and then entered it in the IUCN Species Information Service database (https://sis.iucnsis.org/apps/org.iucn.sis.server/SIS/index.html). Following the supporting information guidelines for the IUCN Red List[67], information was recorded on the geographical distribution, population status and trend, habitats and ecological requirements, use and trade, threats, and research and conservation actions relevant to the species, with supporting sources cited. In addition, a distribution map was produced for each species (see 'Distribution maps' below). The Red List criteria[37] were then applied to assign a Red List category indicating the extinction risk of the species: extinct (EX), extinct in the wild (EW), critically endangered (CR), endangered (EN), vulnerable (VU), near threatened (NT), least concern (LC) or data deficient (DD). Threatened species are those categorized as critically endangered, endangered or vulnerable, and are considered to be facing a high to extremely high risk of extinction in the wild. Species assessed as critically endangered may further be tagged as possibly extinct or possibly extinct in the wild. The former refers to species that are likely to be extinct, but for which there is a small possibility they are still extant, and the latter refers to species that still survive in captivity[61].

Each assessment then underwent two reviews. First, at least one independent scientist familiar with each species reviewed the assessment to ensure the data presented were correct and complete, and that the Red List criteria had been applied appropriately. Once each assessment had passed this first stage of the review, including revision (if necessary), staff from the IUCN Red List Unit reviewed the assessments to ensure that the Red List criteria had been applied appropriately, and the documentation standards had been met. Once each assessment had passed this second stage of the review, again including revision (if necessary), they were considered finalized and set for publication on the IUCN Red List website.

The preliminary Red List assessments of 1,577 freshwater fish species used in this analysis had undergone only the first step of the review process described above at the time of analysis. Of these assessments, 87% (1,370 species) are now published, with only seven species (0.5%) changing Red List category before publication as a result of the second stage of review. We expect the remaining 207 species (13%) with preliminary assessments to have completed the assessment process outlined above by October 2024.

## Taxonomy

For freshwater fishes, we used Eschmeyer's Catalog of Fishes[63] as a taxonomic source, and for odonates, we used the World Odonata List[64]. In both cases, we diverged only to follow well-justified taxonomic standards as recommended by the relevant IUCN SSC Specialist Group. We were not able to revisit new descriptions for regions or families after the end of the original regional sub-projects. Therefore, the final species lists are not fully consistent with any single release of either of the above taxonomic sources.

## Distribution maps

Where data allowed, we produced polygon distribution maps for each species following the IUCN Red List mapping standards[68]. These distribution maps represent the best-available depiction of the historical, present and inferred distribution of a species. They represent the limits of the distribution of a species, indicating that the species probably only occurs within the polygon, but not necessarily everywhere within the polygon. Each polygon was coded according to the species presence, origin and seasonality in the corresponding area. For freshwater species, these polygons were based on river and lake catchments, which are generally accepted as the most appropriate management unit for inland waters[69]. These catchments are delineated using HydroBASINS, a globally standardized hydrological framework[70], with level 8 HydroBASINS as the default resolution (mean sub-basin area of 576 km$^2$). Assessors are recommended to use higher-resolution level HydroBASINS (that is, levels 10 and 12) for species with restricted distributions.

## Threats

All known past, current and future threats to a species were coded using the IUCN Threats Classification Scheme (version 3.3; https://www.iucnredlist.org/resources/threat-classification-scheme).

## Habitat preferences

Where known, species habitats were coded using the IUCN Habitats Classification Scheme (version 3.1; https://www.iucnredlist.org/resources/habitat-classification-scheme). Species were assigned to all habitat classes in which they are known to occur.

## Data limitations

Although we made an extensive effort to complete assessments for all freshwater fishes and odonates, some data gaps remain. These data gaps are discussed below, along with limitations with the published freshwater decapod data.

**Missing species.** As of December 2022, 3,804 freshwater fishes (21% of the formally described species), 169 odonates (3% of the formally described species) and 622 freshwater decapods (19% of the formally described species) were omitted from the study.

For freshwater fishes, the majority of these species were omitted because they were described after the relevant regional sub-projects had already been completed. The earliest sub-projects on the freshwater fishes focused on Pan-Africa[71] (2003–2011; Supplementary Table 1a), and therefore, we would expect the most new descriptions for this region given that the greatest time period has passed. However, a number of regional sub-projects on highly diverse African sub-regions (for example, the Lake Victoria basin[72], the Lake Malawi/Nyasa/Nyassa basin[59] and western Africa[73]) have been completed more recently and will have captured much of this new diversity. In addition, the regions with highest rates of description (that is, South America and Asia)[74,75] have been the focus of more recent regional sub-projects.

The odonates are the most comprehensively assessed of the three freshwater groups with the few omitted species being new descriptions, reflecting the geographical distribution of the order overall.

For freshwater decapods, the majority of omissions were because of new descriptions[65] after the global assessment efforts[38–40] were completed. New descriptions also reflect the geographical distribution of the group overall and have primarily been in Asia, followed by the Neotropics and Afrotropics, for crabs; in Asia, followed by Oceania and Latin America, for shrimps; and in North America, followed by Australia, for crayfishes. Therefore, we do not expect a geographical bias in the results for the freshwater species due to recent descriptions, although we underestimate overall species richness[65].

Recently described species are often not well known and may be rare or occur in very restricted or poorly surveyed areas that are often subject to high levels of human impacts. As such, recent descriptions are more likely to be assessed as data deficient or in a threatened category[76], than to be assessed as least concern. The effect on our analyses is likely to be an underestimate of the number of threatened species and lower surrogacy levels than reported here.

For the freshwater fishes, a secondary reason for the omission of species was that some of the regional sub-projects were still under way at the time of data export in March 2023 (Supplementary Table 1), such that not all species had assessments that had sufficiently progressed through the Red List assessment process to be considered. The omissions for this reason are primarily freshwater fishes native to China (about 850 species), South America (approximately 670 species), India (about 190 species) and the Korean Peninsula (approximately 30 species). With the exception of China, where only approximately 50% of freshwater fishes are assessed, coverage of freshwater fish species in the other countries or regions is considered sufficiently comprehensive (that is, 80% or more) to evaluate the status of the taxonomic group[77,78].

**Data deficient species.** An assessment of data deficient indicates that there is insufficient information available on the distribution and/or population status of a species to make a direct or indirect assessment of its risk of extinction[37]. Eighteen per cent of freshwater fishes (2,634 species), 29% of odonates (1,830 species) and 39% of freshwater decapods (1,042 species) were assessed as data deficient.

**Spatial data.** Although we made extensive efforts to map the current known distribution of each species, this is missing or incomplete for some species. Polygon map availability for all freshwater species was as follows: freshwater decapods (94%), freshwater fishes (95%) and odonates (77%). Polygon map availability for threatened freshwater species was as follows: freshwater decapods (89%), freshwater fishes (96%) and odonates (87%). Polygon map availability for data deficient freshwater species was as follows: freshwater decapods (93%), freshwater fishes (92%) and odonates (77%). Polygon map availability for threatened tetrapod species was as follows: amphibians (100%), birds (100%), mammals (100%) and reptiles (96%). Species missing maps generally had older Red List assessments where the supporting data requirements were lower or only had non-polygon spatial data (that is, point locality data).

It should be noted that species occurrence is unlikely to be spread evenly or entirely throughout the area depicted in species distribution maps, for example, with gaps expected in areas without suitable habitat.

**Time span of assessments.** Reassessment of species on the IUCN Red List is recommended every 10 years[79], with assessments older than this tagged as 'needs updating'. The freshwater fish assessments were completed between 1996 and 2023, with 4,834 assessments (33%) completed before 2013. The odonate assessments were completed between 1996 and 2021, with 1,663 (27%) completed before 2013. The freshwater decapod assessments were completed between 1996 and 2022, with 2,482 (94%) completed before 2013. Approximately the same proportion of freshwater fishes were assessed as threatened in the pre-2013 (27%) and post-2013 (26%) datasets. However, a higher proportion of

odonates and freshwater decapods were assessed as threatened in the post-2013 dataset (17% and 42%, respectively) than the pre-2013 dataset (12% and 29%, respectively). This could be because the later assessments focused more on the tropical Americas and Asia, where species have smaller ranges on average than in Africa and the Holarctic, which is likely to lead to a higher percentage threatened. Given the continuing decline of biodiversity globally[6], it is probable that species with assessments older than 10 years are more likely to be assessed in a higher threat category today, indicating a potential underestimation of the extinction risk in these groups.

### Analyses

**Proportion of species threatened with extinction.** To capture the uncertainty in the proportion of species threatened with extinction that is introduced by data deficient species, we report three values for percentage threatened as follows.

We used the following formula as the midpoint of the proportion of species threatened with extinction:

$$(EW + CR + EN + VU)/(N - EX - DD)$$

where EW, CR, EN, VU, EX and DD are the number of species in each corresponding Red List category, and $N$ is the total number of species assessed. This formula assumes data deficient species are equally threatened as data-sufficient species. This value is considered the best estimate of extinction risk[78].

We used the following formula to calculate the lower estimate of the proportion of species threatened with extinction:

$$(EW + CR + EN + VU)/(N - EX)$$

where EW, CR, EN, VU and EX are the number of species in each corresponding Red List category, and $N$ is the total number of species assessed. This formula assumes data deficient species are not threatened.

Finally, we used the following formula to calculate the upper estimate of the proportion of species threatened with extinction:

$$(EW + CR + EN + VU + DD)/(N - EX)$$

where EW, CR, EN, VU, DD and EX are the number of species in each corresponding Red List category, and $N$ is the total number of species assessed. This formula assumes that all data deficient species are threatened.

**Threats.** We analysed threats as classified under the IUCN Threats Classification Scheme. Multiple threats can affect a single species. It is possible to assign codes for scope and severity to each threat. However, this is optional[67] and not available for the majority of freshwater species. Therefore, we did not use scope and severity to distinguish the relative importance of threats to individual species, and all coded threats were included in the analysis. We recommend future reassessments to include these codes, such that major threats can be distinguished from trivial threats in analyses based, for example, on the proportion of the population affected.

The threats analysis presented here focuses on threatened species, plus on extinct and extinct in the wild freshwater species. Threats data availability for threatened species for each taxonomic group was as follows: freshwater decapods (97%), freshwater fishes (98%), odonates (97%), amphibians (100%), birds (100%), mammals (99%) and reptiles (97%). Eighty-two per cent of extinct and extinct in the wild freshwater species had threats coded. Species without threats data either faced no known major threats, faced unknown threats or had no threats data coded because they have older Red List assessments where the supporting data requirements were lower.

**Habitats.** We first analysed habitat use at the highest level of the IUCN Habitats Classification Scheme, focusing on threatened species. In addition, for freshwater species only, we analysed habitat use within habitat code 5 'Wetlands (inland)', focusing on all species, and then threatened, extinct in the wild and extinct species.

Habitat data availability for threatened species for each taxonomic group was as follows: freshwater decapods (99%), freshwater fishes (99%), odonates (97%), amphibians (100%), birds (100%), mammals (99%) and reptiles (97%). Wetland-specific habitat data availability for all species for each freshwater taxonomic group was as follows: freshwater decapods (100%), freshwater fishes (99%) and odonates (99%). All extinct and extinct in the wild freshwater species had habitats coded. Species without habitat data either had habitats coded as 'unknown' or had no habitat data coded because they have older Red List assessments in which the supporting data requirements were lower.

**Statistical tests.** We used chi-squared tests to assess whether any threats were recorded more for extinct species than would be expected based on threats recorded for threatened species, whether any habitats were recorded more for threatened species than would be expected based on habitats recorded for all species, and whether any habitats were recorded more for extinct species than would be expected based on habitats recorded for threatened species.

**Spatial analyses.** Analyses of the distribution maps used polygons coded with the following presence, origin and seasonality values (as defined in the IUCN Red List mapping standards[68]):
- Presence denotes extant (code 1), probably extant (code 2) or possibly extinct (code 4)
- Origin refers to native (code 1), reintroduced (code 2) or assisted colonization (code 6)
- Seasonality indicates resident (code 1), breeding (code 2), non-breeding (code 3) or passage (code 4)

In addition, for the surrogacy analyses, we excluded ranges coded as presence code 4 (possibly extinct).

All spatial mapping and subsequent analyses were conducted on a global 0.5 × 0.5 latitude–longitude grid (approximately 50-km resolution; WGS84). We converted all polygon range maps (including HydroBASIN-based maps) tagged with presence, origin and seasonality codes as described above to these grids to have a consistent format for analysis across all taxonomic groups. We mapped the distribution of species as a count of the number of species ranges overlapping each grid cell.

**Surrogacy estimation analyses within two conservation strategies.** To evaluate the degree to which conservation of threatened amphibians, birds, mammals and reptiles (individually or combined) serve as surrogates for conservation of threatened freshwater decapods, fishes and odonates (individually or combined), we calculated the SAI of surrogate effectiveness[33]. A surrogate is selected as a representative of conservation planning, simplifying the process of monitoring and conserving biodiversity. Its effectiveness hinges on how well the surrogate can reflect the presence, abundance and diversity of species in a given area. Here we used species accumulation curves to measure this effectiveness, by comparing the species accumulation curves of surrogates with those of the target group.

We performed the analyses on two main global conservation strategies: (a) maximizing rarity-weighted richness (that is, the aggregate importance of each grid cell to the species occurring there) of threatened species, and (b) maximizing inclusion of the most range-restricted threatened species. The first strategy prioritizes areas containing many threatened species with highly restricted ranges globally, whereas the second prioritizes essential areas for the most globally range-restricted threatened species.

We implemented both conservation strategies within the spatial conservation planning software Zonation[80] and the R[81] package 'zonator'[82], using the additive benefit function (ABF) and the core-area zonation (CAZ) algorithms for strategy a and strategy b, respectively. The algorithm for the ABF (strategy a) focuses on ranking areas by the sum of the proportion of the overall range size of all species found within a specific grid cell (that is, a quantity similar to weighted species endemism and endemism richness). The grid cells that contain many species occurring exclusively in that cell or in only a few other cells are given the highest priority. In the CAZ algorithm (strategy b), areas are prioritized based on the maximum proportion of the global range size of all species within a specific grid cell. The algorithm assigns the highest priority to cells that contain the greatest proportions of the ranges of the most range-restricted species.

We estimated optimal, surrogate and random curves based on multiple target species-surrogate species combinations. We used 100 sets of random terrestrial grid-cell sequences to generate 95% confidence intervals around a median random curve. We ran five iterations of each spatial prioritization algorithm for each taxonomic group, and optimal and surrogate curves were summarized using the median and 95% confidence intervals across the five iterations.

We derived the SAI of surrogate effectiveness[83], which quantifies the rate of inclusion of target biodiversity units across areas selected optimally based on the targets themselves, based on surrogate diversity, or at random, as

$$(s - r)/(o - r)$$

where $s$ is the area under the surrogate curve, $r$ is the area under the random curve, and $o$ is the area under the optimal curve. If SAI = 1, the optimal and the surrogate curves coincide (perfect surrogacy); if SAI is between 1 and 0, the surrogate curve is above the random curve (positive surrogacy); if SAI = 0, the surrogate and random curves are the same (no surrogacy); and if SAI < 0, the surrogate curve is below the random curve (negative surrogacy). We used the following descriptors to define SAI performance: 0.01–0.19 as very poor, 0.20–0.39 as poor, 0.40–0.59 as reasonable, 0.60–0.79 as good, and 0.80–0.99 as very good. It should be noted that if SAI = 0.5, for example, this does not mean that 50% of targets are represented and 50% of targets are not represented. For each SAI, we reported the median and 95% confidence intervals based on the five target and surrogate curve iterations and 100 random curve iterations.

In addition, we evaluated whether prioritizing for two widely used hydrological variables (water stress as a measure of water quantity, and eutrophication (nitrate–nitrite) as a measure of water quality) are effective surrogacy strategies for conservation of threatened freshwater species. We used the SAI to evaluate the ability of both variables to identify areas that most efficiently represent threatened freshwater species, again harnessing strategies for both maximizing rarity-weighted richness (ABF) and maximizing inclusion range-restricted species (CAZ). Once again, we used Zonation[80] to generate the complementarity-based ranking of conservation values of the target, with the respective algorithms, over the landscape of interest. To generate the rank order, we used (1) the baseline water stress layer from the Aqueduct Water Risk Atlas, which measures the ratio of total water demand (for example, domestic, industrial, irrigation, and livestock consumptive and non-consumptive uses) to available renewable surface and groundwater supplies[84,85], and (2) the baseline nitrogen layer from the World Bank catalogue, which provides global predictions of nitrate–nitrite levels[86]. The water stress layer was considered a proxy of a baseline level of water demand compared with available renewable water and groundwater, as used in setting science-based targets for freshwater[8]. Nitrogen levels in water around the world are highly correlated with population density, sanitation practices and agricultural activities. Here the nitrogen layer was predicted globally and provides valuable

information about nitrogen concentrations in areas where no previous observations have been made.

We rasterized the baseline water stress and the nitrogen layers to a 0.5 × 0.5 latitude–longitude grids (approximately 50-km resolution; WGS84) to match the species rasters. For the water stress analysis, we excluded cells with missing water stress data across the world's land (12% of cells excluded). We found that 44% of the world's cells with water stress data had no threatened freshwater species, but these cells were still included in the analysis. For the nitrogen levels analysis, we excluded cells missing nitrogen data across the world's land, which accounted for 16% of the cells. Among the remaining cells with nitrogen data, 52% had no threatened freshwater species, but again these were retained in the analysis. Before constructing the curves, we organized sites (grid cells) in the species matrix from those with high abiotic values to low abiotic values for ranking cells. We used 100 sets of random terrestrial grid-cell sequences to generate 95% confidence intervals around a median random curve. We generated five random terrestrial grid cell sequences for constructing the surrogate curves, so we randomly changed the rank order only between those cells that have the same values.

## Reporting summary

Further information on research design is available in the Nature Portfolio Reporting Summary linked to this article.

## Data availability

Taxonomic data for freshwater fishes are available from Eschmeyer's Catalog of Fishes (http://researcharchive.calacademy.org/research/ichthyology/catalog/fishcatmain.asp) and for odonates from the World Odonata List (https://www.pugetsound.edu/slater-museum-natural-history-0/biodiversity-resources/insects/dragonflies/world-odonata-list). All IUCN Red List assessment data are publicly available on the IUCN Red List of Threatened Species website (www.iucnredlist.org). Occasionally, where a species may be under threat from over-collection, sensitive spatial data are not publicly available. All tabular and spatial data used in the analyses ('One-quarter of freshwater fauna threatened with extinction') are available (https://www.iucnredlist.org/resources/data-repository). Baseline water stress data ('Aqueduct water stress projections data') are available from the Aqueduct Water Risk Atlas (https://www.wri.org/data/aqueduct-water-stress-projections-data). Baseline nitrogen data ('Global − nitrate–nitrite in surface water') are available from the World Bank catalogue (https://datacatalog.worldbank.org/search/dataset/0038385/Global−Nitrate-nitrite-in-Surface-Water). Source data are provided with this paper.

## Code availability

The code used for the surrogacy analyses is available at Zenodo[87] (https://doi.org/10.5281/zenodo.10286099). No code was used for the chi-squared tests, which were performed in Microsoft Excel.

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

Steering Committee. *IUCN Red List* https://nc.iucnredlist.org/redlist/content/attachment_files/RL_Standards_Consistency.pdf (2013).

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

**Acknowledgements** We thank the IUCN Red List assessors (Supplementary Note 1); N. Bogutskaya, M. Dagou Diop, M. Entsua-Mensah, M. Kretschmar, T. Lowe, A. McIvor and W. Vishwanath for their contributions to coordinating the IUCN Red List assessment efforts for freshwater fishes; and J. Hart for assistance in processing the nitrogen data layer for the surrogacy analysis. We acknowledge funding and support for the global assessment efforts for freshwater fishes and odonates from Al-Farabi Kazakh National University; Asian Development Bank; Ashoka Trust for Research in Ecology and the Environment; Auckland Zoo; California Academy of Sciences; Câmara Municipal de Vila do Conde; Center for Species Survival, New Mexico BioPark Society; Comitato Italiano IUCN; Comité Français de l'UICN; Conservation International; Critical Ecosystem Partnership Fund; Department of Fisheries, Malawi; Dirección General de Diversidad Biológica del Ministerio del Ambiente de Perú; Dutch Ministry of Foreign Affairs; Environment Agency Abu Dhabi; European Union; Federazione Italiana dei Parchi e delle Riserve Naturali (Federparchi); Fonds Pacifique; Global Center for Species Survival, Indianapolis Zoo; Instituto Chico Mendes de Conservação da Biodiversidade; Instituto de Investigación de Recursos Biológicos von Humboldt; IBAT; International Institute for Geo-Information Science and Earth Observation; IUCN; IUCN's Bureau Regional de l'Afrique Centrale et Ouest; IUCN Moroccan National Committee; IUCN National Committee of the Netherlands; IUCN SSC; IUCN Tunisian National Committee; IUCN Water and Nature Initiative; John D. and Catherine T. MacArthur Foundation; JRS Biodiversity Foundation; Junta de Andalucia; Lee Kong Chian Natural History Museum, National University of Singapore; Leibniz-Institute of Freshwater Ecology and Inland Fisheries; Mandai Wildlife Group; MAVA Foundation; Ministero dell'Ambiente e della Tutela del Territorio e del Mare; Missouri Botanical Garden, Madagascar; Monash University; Museum and Art Gallery of the Northern Territory; Muséum National d'Histoire Naturelle; National Institute of Biological Resources; National Museums of Kenya; NatureServe; North of England Zoological Society; Office Pour les Insectes et leur Environnement; Pontificia Universidad Javeriana de Colombia; Research Centre for Biodiversity and Genetic Resources of Porto University; Rufford Foundation; Sapienza Università di Roma; Senckenberg Research Institute and Natural History Museum; Sociedade de Odonatología Latinoamericana; Société française d'Odonatologie; Society of Entrepreneurs and Ecology Foundation; South Africa Institute for Aquatic Biodiversity; South Africa National Biodiversity Institute; Spanish Agency for International Cooperation Development; Spanish Ministry of Environment; Synchronicity Earth; Tanzania Fisheries Research Institute; Toyota Motor Corporation through the IUCN-Toyota Red List Partnership; Uganda Coalition for Sustainable Development; Ugandan National Wetlands Programme; University of Burundi; University of Canberra; Universidad de Los Andes; Wetlands International; WildFish; WorldFish; Yayasan Bumi Sawerigading; Zoo Outreach Organisation; and Zoological Society of London. We acknowledge funding for this analysis and manuscript from the Regina Bauer Frankenberg Foundation; the Global Environment Facility (GEF); and the support of the IUCN GEF Project Agency. The views expressed in this publication do not necessarily reflect those of IUCN. The designation of geographical entities in this paper, and the presentation of the material, do not imply the expression of any opinion whatsoever on the part of IUCN concerning the legal status of any country, territory or area, or of its authorities, or concerning the delimitation of its frontiers or boundaries.

**Author contributions** C.A.S. conceived the original idea and curated the data. C.A.S., E.F., R.R.J. and N.B.W.M. ran the analyses with assistance from G.R. N.A.C. and W.R.T.D. secured the principal funding for the analysis and manuscript. C.A.S. and R.R.J. wrote the first draft. C.A.S., E.F., R.R.J., N.B.W.M., G.R., M.B., T.M.B., T.C.-B., N.A.C., I.H., M.H., R.J., K.G.S., J.-C.V., J.C.A., D.J.A., G.R.A., V.B., J.-P.B., S.F.C., P.C., V.C., L.C., K.A.C., N.C., A.C., J.D., A.G.D., S.D.G., G.D.K., K.-D.B.D., R.A.D., J.F., N.G., J.G., A.G., C.G., M.J.G., M.I.G., A.E.R.G., M.P.H., G.A.H., C.H.-T., L.H., R.A.H., R.W.J., D.J.B., V.J.K., B.K.K., J.K., M.K., P.A.L., H.K.L., M.L., F.L., A.L., T.J.L., L.M.-T., S.M., H.H.N., C.N., A.F.P.-N., C.P., H.E.P., C.N.M.P., C.M.P., R.R., P.S.R., T.R., R.E.R., C.L.R., J.A.S., P.H.S., M.R.S., J.S., M.L.J.S., H.H.T., Y.T., E.B.T., M.F.T., A.G.T., Y.T., D.T., K.W., J.R.S.W., E.G.E.W., E.Z. and W.R.T.D. all reviewed and edited the manuscript.

**Competing interests** G.R., M.L. and L.M.-T. are affiliated with commercial companies (Elimia, Fish Fondler Pty Ltd. and Mott MacDonald, respectively), but their contributions to this paper were made in a voluntary capacity. The other authors declare no competing interests.

**Additional information**
**Correspondence and requests for materials** should be addressed to Catherine A. Sayer.

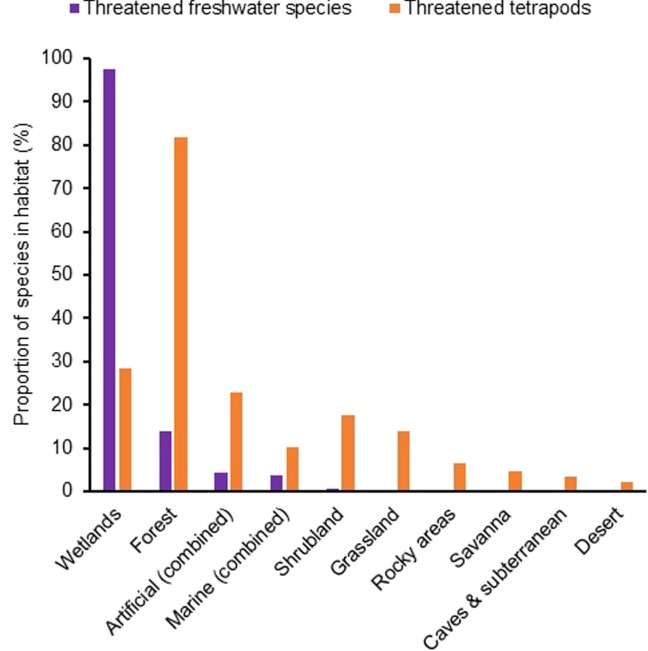

**Extended Data Fig. 1 | Habitats used by threatened freshwater species (decapod crustaceans, fishes, and odonates; combined) and threatened tetrapods.** Threatened species include those assessed as Critically Endangered, Endangered, or Vulnerable (including those flagged as Possibly Extinct and Possibly Extinct in the Wild). Habitats are not mutually exclusive. Habitats are coded following the IUCN Habitats Classification Scheme (version 3.1) and combined for presentation as follows (value of highest hierarchical level is indicated, all subsequent levels are included): wetlands (5); forest (1); artificial (combined) (14, 15); marine (combined) (9, 10, 11, 12, 13); shrubland (3); grassland (4); rocky areas (6); savanna (2); caves & subterranean (7); and desert (8). The following habitats are not shown: introduced vegetation (16); other (17); and unknown (18). Number of species: threatened freshwater species n = 4,236; and threatened tetrapods n = 7,108.

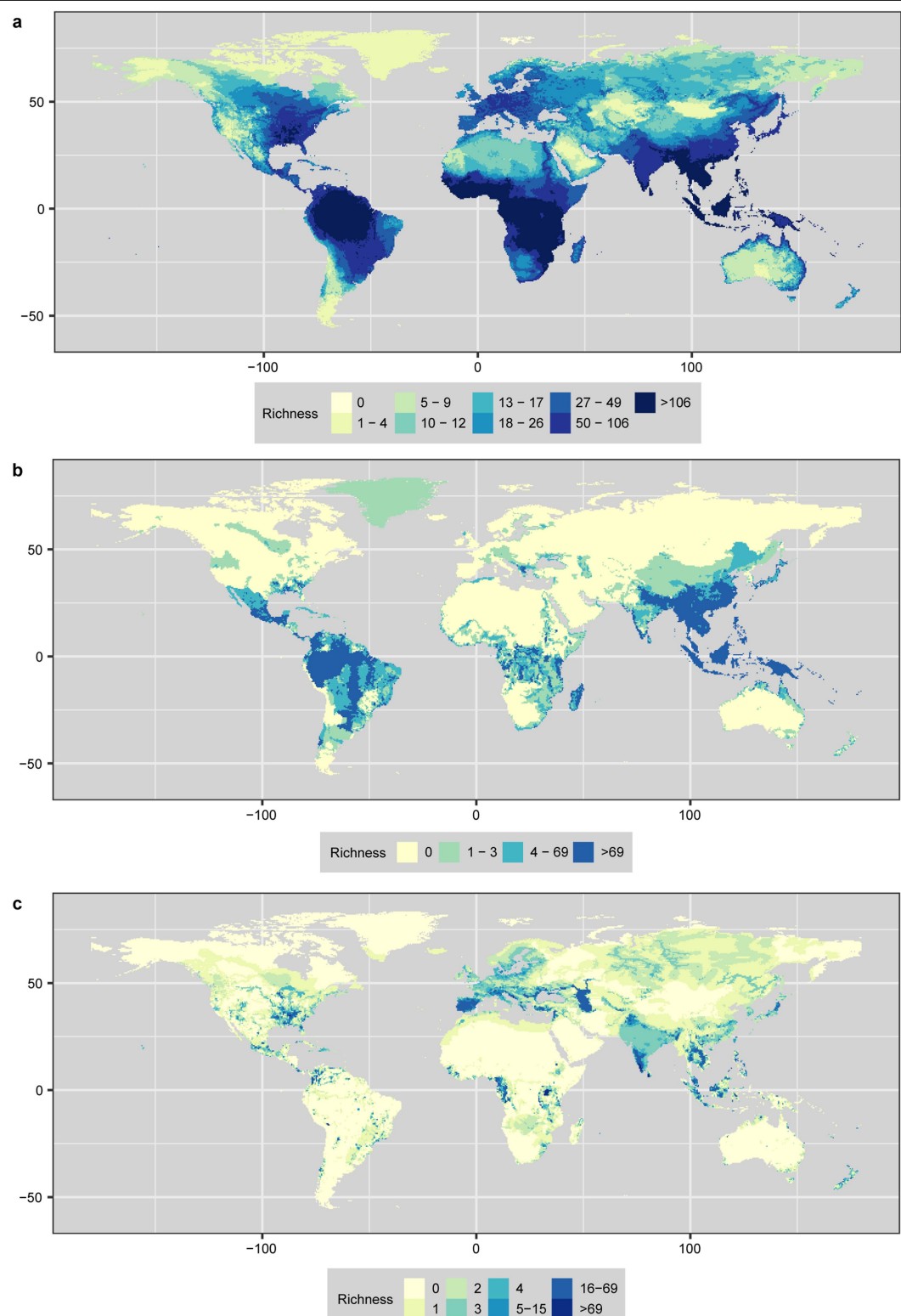

**Extended Data Fig. 2 | Absolute richness of freshwater species for a) all freshwater species; b) Data Deficient freshwater species; and c) threatened freshwater species (excluding Critically Endangered (Possibly Extinct) and Critically Endangered (Possibly Extinct in the Wild) species).** The following distributions are included: Presence = Extant, Probably Extant, or Possibly Extinct; Origin = Native, Reintroduced, or Assisted Colonisation; and Seasonality = Resident, Breeding, Non-breeding, or Passage. The value of each cell is calculated as the count of species with a mapped distribution overlapping the cell. Richness shown using a 0.5 × 0.5 latitude-longitude grid and WGS84. World Bank Official Boundaries (licensed under a Creative Commons licence CC BY 4.0) were used as the base map.

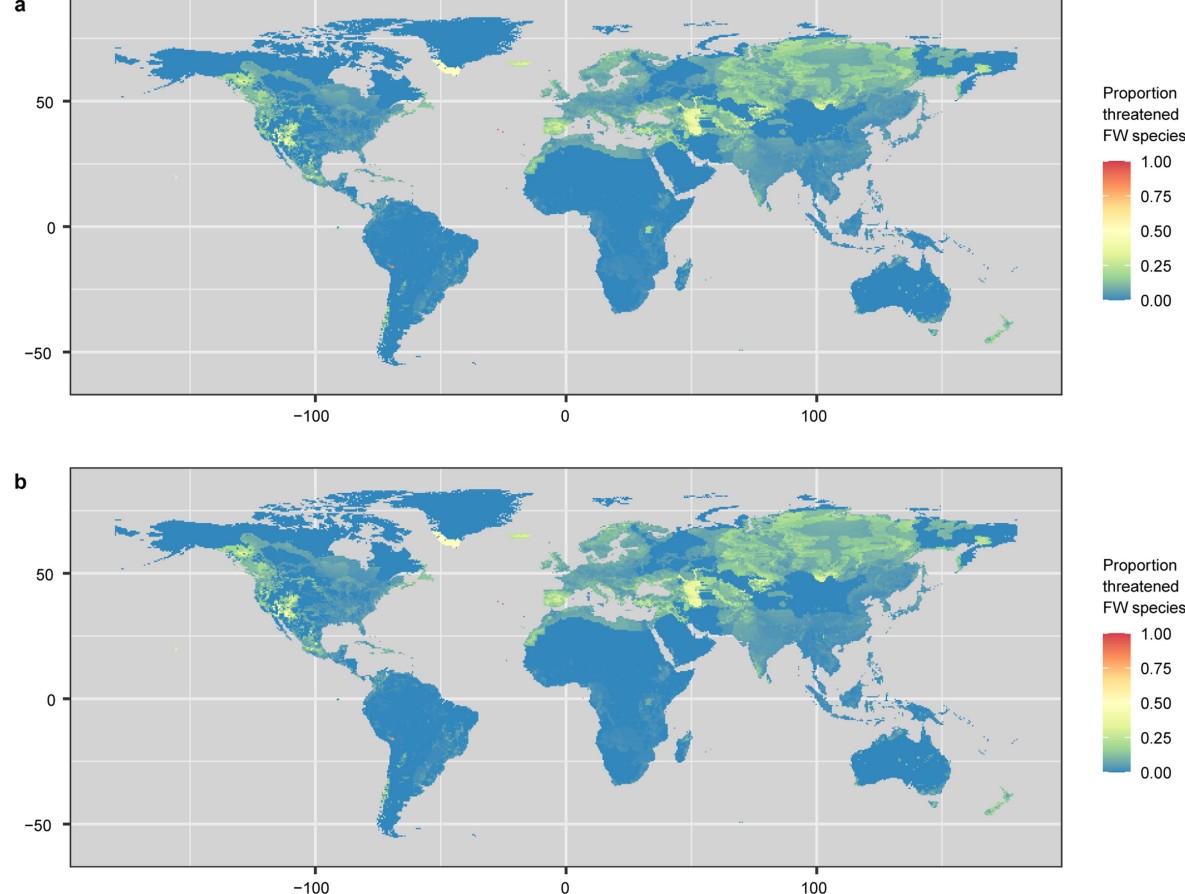

**Extended Data Fig. 3 | Proportional richness of threatened freshwater species a) including Critically Endangered (Possibly Extinct) and Critically Endangered (Possibly Extinct in the Wild) species; and b) excluding Critically Endangered (Possibly Extinct) and Critically Endangered (Possibly Extinct in the Wild) species.** The following distributions are included: Presence = Extant, Probably Extant, or Possibly Extinct; Origin = Native, Reintroduced, or Assisted Colonisation; and Seasonality = Resident, Breeding, Non-breeding, or Passage. The value of each cell is calculated by its absolute threatened species richness divided by its absolute species richness. Richness shown using a 0.5 × 0.5 latitude-longitude grid and WGS84. World Bank Official Boundaries (licensed under a Creative Commons licence CC BY 4.0) were used as the base map.

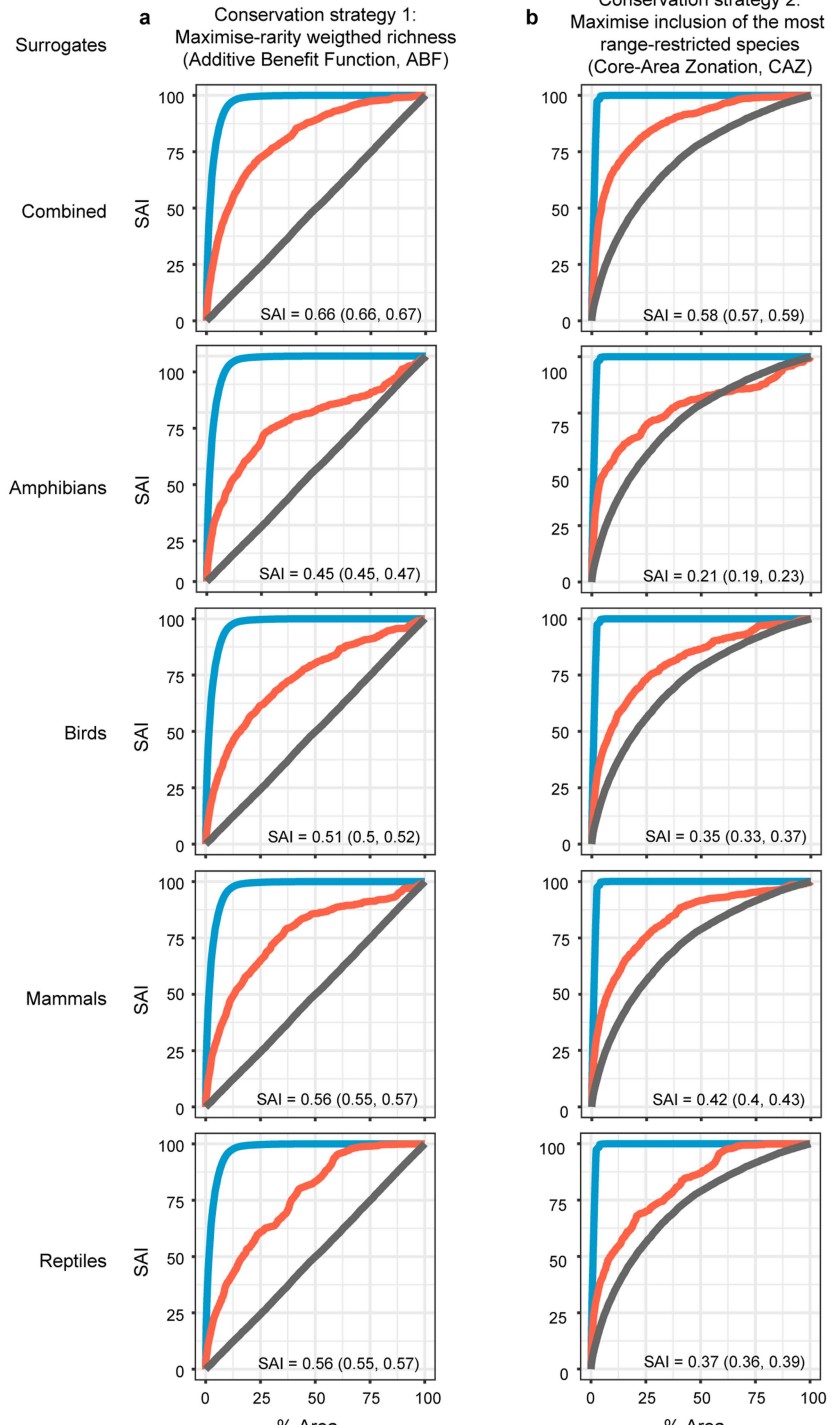

**Extended Data Fig. 4 | Species Accumulation Index (SAI) values and curves from the surrogacy analysis, indicating the effectiveness of tetrapods (combined or individually) as surrogates for freshwater species (combined) targets.** Values and curves are shown for two alternative conservation strategies: **a)** maximises rarity-weighted richness, and **b)** maximises inclusion of the most range-restricted species. See Methods for a full explanation of each strategy. Surrogate effectiveness is measured using the Species Accumulation Index (SAI): values range from −∞ to 1, with 1 indicating perfect surrogacy, values between 1 and 0 indicating positive surrogacy, 0 indicating no surrogacy, values less than 0 indicating negative surrogacy. In each panel, median SAI values are provided, with lower and upper confidence intervals in brackets. Blue lines are the optimal curves (accumulation of target diversity based on target priority areas); red lines are the surrogate curves (accumulation of target diversity based on surrogate priority areas); and grey lines are the random curves (accumulation of target diversity based on random selection of areas). Confidence intervals (95%, based on 100 randomisations) are shown in lighter shading around curves; most are too small to be visible.

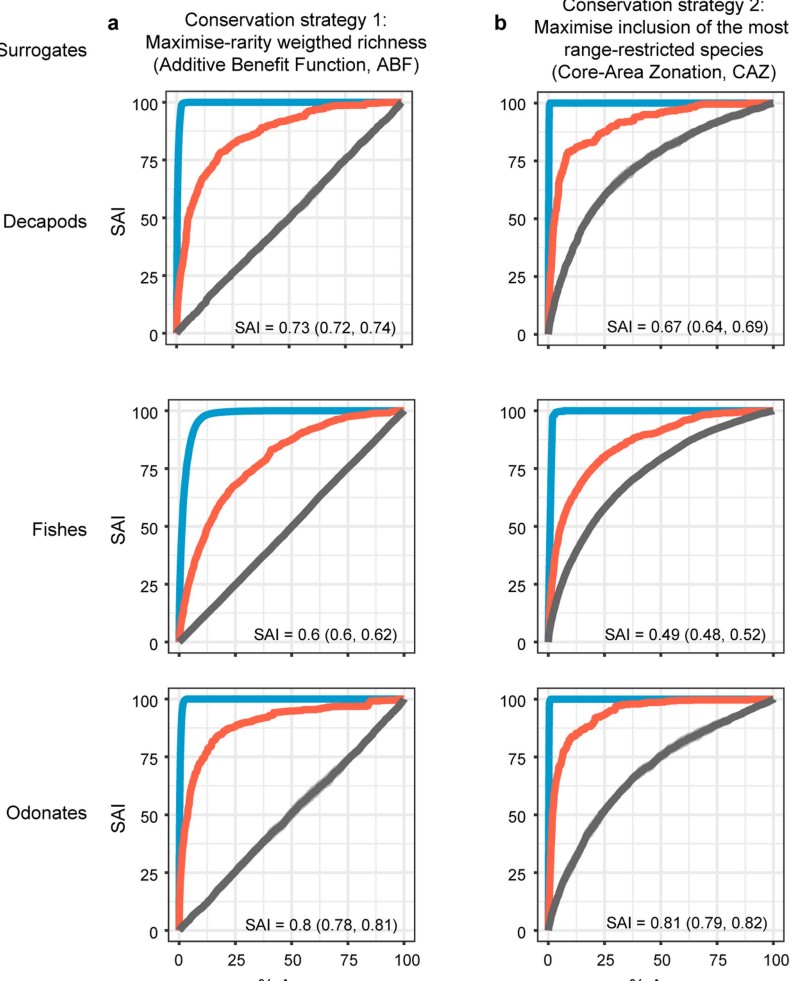

**Extended Data Fig. 5 | Species Accumulation Index (SAI) values and curves from the surrogacy analysis, indicating the effectiveness of tetrapods (combined) surrogates for freshwater species (individually) targets.** Values and curves are shown for two alternative conservation strategies: **a)** maximises rarity-weighted richness, and **b)** maximises inclusion of the most range-restricted species. See Methods for a full explanation of each strategy. Surrogate effectiveness is measured using the Species Accumulation Index (SAI): values range from −∞ to 1, with 1 indicating perfect surrogacy, values between 1 and 0 indicating positive surrogacy, 0 indicating no surrogacy, and values less than 0 indicating negative surrogacy. In each panel, median SAI values are provided, with lower and upper confidence intervals in brackets. Blue lines are the optimal curves (accumulation of target diversity based on target priority areas); red lines are the surrogate curves (accumulation of target diversity based on surrogate priority areas); and grey lines are the random curves (accumulation of target diversity based on random selection of areas). Confidence intervals (95%, based on 100 randomisations) are shown in lighter shading around curves; most are too small to be visible.

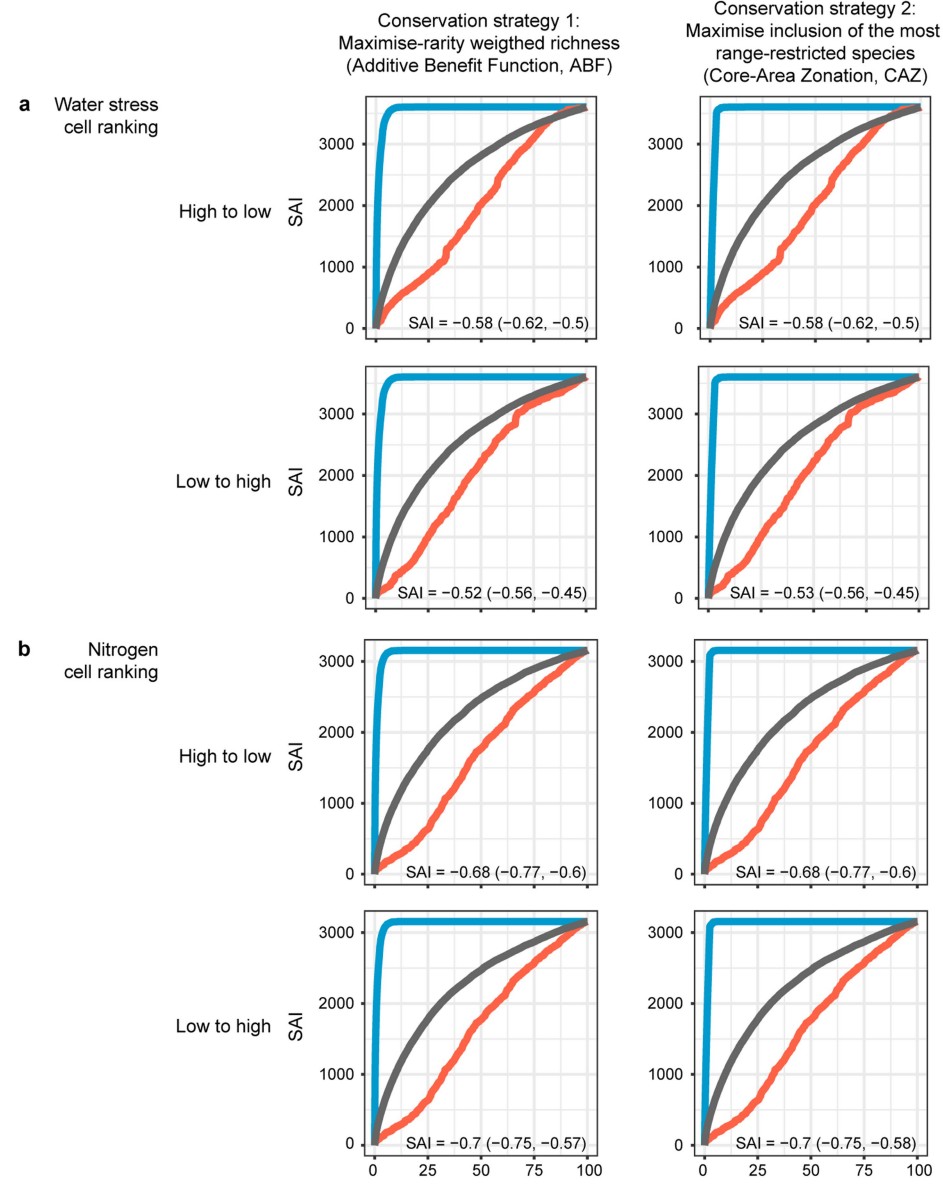

**Extended Data Fig. 6 | Species Accumulation Index (SAI) values and curves from the surrogacy analysis, indicating the effectiveness of two abiotic factors as surrogates for freshwater species (combined) targets: a) water stress, and b) Nitrogen (as a proxy of eutrophication).** Values and curves are shown for two alternative conservation strategies: 1) maximises rarity-weighted richness, and 2) maximises inclusion of the most range-restricted species. See Methods for a full explanation of each strategies and values of water stress. Surrogate effectiveness is measured using the Species Accumulation Index (SAI): values range from −∞ to 1, with 1 indicating perfect surrogacy, values between 1 and 0 indicating positive surrogacy, 0 indicating no surrogacy, and values less than 0 indicating negative surrogacy. In each panel, median SAI values are provided, with lower and upper confidence intervals in brackets. Blue lines are the optimal curves (accumulation of target diversity based on target priority areas); red lines are the surrogate curves (accumulation of target diversity based on surrogate priority areas); and grey lines are the random curves (accumulation of target diversity based on random selection of areas). Confidence intervals (95%, based on 100 randomisations) shown in lighter shading around curves; most are too small to be visible.

**Extended Data Table 1 | Summary of Red List Categories for all freshwater taxonomic groups**

| Red List Category | Freshwater taxonomic group | | | |
|---|---|---|---|---|
| | Decapods | Fishes | Odonates | Total |
| EX | 6 | 82 | 1 | 89 |
| EW | 0 | 11 | 0 | 11 |
| CR | 116 | 737 | 96 | 949 |
| CR (PE) | 18 | 141 | 19 | 178 |
| CR (PEW) | 1 | 8 | 0 | 9 |
| EN | 144 | 1,154 | 311 | 1,609 |
| VU | 226 | 1,209 | 301 | 1,736 |
| NT | 70 | 686 | 237 | 993 |
| LC | 1,041 | 8,115 | 3,447 | 12,603 |
| DD | 1,042 | 2,634 | 1,830 | 5,506 |
| Total | 2,645 | 14,628 | 6,223 | 23,496 |
| % DD | 39% | 18% | 29% | 23% |
| % threatened (best estimate) | 30% | 26% | 16% | 24% |
| % threatened (lower estimate) | 18% | 21% | 11% | 18% |
| % threatened (upper estimate) | 58% | 40% | 41% | 42% |

Red List Categories are as follows: EX, Extinct; EW, Extinct in the Wild; CR, Critically Endangered; EN, Endangered; VU, Vulnerable; NT, Near Threatened; DD, Data Deficient; and LC, Least Concern. Possibly Extinct (PE) and Possibly Extinct in the Wild (PEW) are tags added to the CR category. See Methods for calculation of the best, lower, and upper estimates of the proportion threatened.

**Extended Data Table 2 | Proportion of threatened versus extinct freshwater species impacted by threats**

| Threat | a | | | b | | |
|--------|---|---|---|---|---|---|
| | **Proportion freshwater species impacted** | | | **Proportion freshwater species impacted** | | |
| | Threatened (incl. PE + PEW) | Extinct (excl. PE + PEW) | *p* | Threatened (excl. PE + PEW) | Extinct (incl. PE + PEW) | *p* |
| Pollution | 0.54 | 0.52 | 0.961 | 0.53 | 0.63 | 0.005* |
| Dams & water management | 0.39 | 0.63 | 2.8E-05* | 0.39 | 0.38 | 0.933 |
| Agriculture | 0.37 | 0.06 | 5.9E-08† | 0.37 | 0.18 | 2.3E-10† |
| Invasive species & disease | 0.28 | 0.55 | 3.7E-07* | 0.27 | 0.50 | 5.0E-16* |
| Logging | 0.25 | 0.04 | 4.1E-05† | 0.26 | 0.08 | 1.3E-10† |
| Urban development | 0.23 | 0.07 | 0.004† | 0.23 | 0.14 | 0.003† |
| Hunting & fishing | 0.21 | 0.37 | 0.003* | 0.20 | 0.42 | 2.2E-17* |
| Energy production & mining | 0.18 | 0.01 | 4.0E-04† | 0.19 | 0.04 | 6.1E-09† |
| Climate change & severe weather | 0.18 | 0.05 | 0.010† | 0.18 | 0.06 | 3.1E-06† |
| Human intrusions & disturbance | 0.08 | 0.02 | 0.212 | 0.08 | 0.04 | 0.104 |
| Other ecosystem modifications | 0.06 | 0.04 | 0.582 | 0.07 | 0.04 | 0.298 |
| Transportation | 0.05 | 0.01 | 0.252 | 0.06 | 0.01 | 0.008† |
| Fire & fire suppression | 0.05 | 0.00 | 0.109 | 0.05 | 0.02 | 0.026† |
| Problematic native species | 0.04 | 0.02 | 0.679 | 0.03 | 0.19 | 3.3E-40* |
| Aquaculture | 0.02 | 0.00 | 0.411 | 0.02 | 0.01 | 0.295 |
| Geological events | 0.01 | 0.00 | 0.653 | 0.01 | 0.01 | 0.971 |

Proportions and p values for differences as tested through two-sided chi-squared tests are given for **a)** inclusion of Critically Endangered (Possibly Extinct) and Critically Endangered (Possibly Extinct in the Wild) species as extant and threatened (because their extinction has not been confirmed), and **b)** inclusion of Critically Endangered (Possibly Extinct) and Critically Endangered (Possibly Extinct in the Wild) species as extinct (because this is expected to be their true status). Otherwise, threatened species include those assessed as Critically Endangered, Endangered, or Vulnerable. Extinct freshwater species include those assessed as Extinct or Extinct in the Wild. p values marked * indicate the threat is impacting more extinct species than expected based on threatened species. p values marked † indicate the threat is impacting fewer extinct species than expected based on threatened species. Degrees of freedom = 1.

**Extended Data Table 3 | Frequencies of pairs of threats leading to extinction of freshwater species**

| Threat | Agriculture | Climate change & severe weather | Dams & water management | Energy production & mining | Human intrusions & disturbance | Hunting & fishing | Invasive species & disease | Logging | Other ecosystem modifications | Pollution | Problematic native species | Transportation | Urban development |
|---|---|---|---|---|---|---|---|---|---|---|---|---|---|
| Agriculture | | 1 | 1 | | 1 | 1 | 3 | 2 | | 3 | | | 2 |
| Climate change & severe weather | | | 2 | | | | 3 | | | 2 | | | |
| Dams & water management | | | | 1 | 2 | 20 | 28 | | 2 | 30 | 1 | 1 | 1 |
| Energy production & mining | | | | | 1 | | | | | | | | |
| Human intrusions & disturbance | | | | | | 2 | | | 2 | | | 1 | |
| Hunting & fishing | | | | | | | 21 | 2 | | 21 | | 1 | 2 |
| Invasive species & disease | | | | | | | | 2 | 1 | 23 | 1 | | 3 |
| Logging | | | | | | | | | | 1 | | | 1 |
| Other ecosystem modifications | | | | | | | | | | 1 | | | 1 |
| Pollution | | | | | | | | | | | 1 | 1 | 2 |
| Problematic native species | | | | | | | | | | | | | 1 |
| Transportation | | | | | | | | | | | | | |
| Urban development | | | | | | | | | | | | | |

Extinct freshwater species include those assessed as Extinct or Extinct in the Wild. Counts indicate the number of extinct freshwater species considered to be impacted by both of the listed threats. Darker cells indicate a greater number of species impacted by both threats.

**Extended Data Table 4 | Proportion of threatened versus extinct freshwater species using each habitat**

| Habitat | a Proportion freshwater species using habitat | | p | b Proportion freshwater species using habitat | | p |
|---|---|---|---|---|---|---|
| | Threatened (incl. PE + PEW) | Extinct (excl. PE + PEW) | | Threatened (excl. PE + PEW) | Extinct (incl. PE + PEW) | |
| Permanent rivers | 0.71 | 0.37 | 1.7E-12† | 0.73 | 0.38 | 2.3E-39† |
| Permanent lakes | 0.18 | 0.48 | 1.1E-13* | 0.17 | 0.47 | 2.6E-39* |
| Seasonal rivers | 0.10 | 0.00 | 0.003† | 0.11 | 0.03 | 1.0E-04† |
| Permanent pools | 0.09 | 0.03 | 0.087 | 0.10 | 0.03 | 7.0E-04† |
| Bogs, marshes, etc. | 0.08 | 0.04 | 0.329 | 0.08 | 0.03 | 0.012† |
| Seasonal pools | 0.08 | 0.00 | 0.014† | 0.08 | 0.03 | 0.005† |
| Springs & oases | 0.06 | 0.19 | 8.4E-08* | 0.06 | 0.10 | 0.008* |
| Karst | 0.05 | 0.00 | 0.067 | 0.05 | 0.02 | 0.105 |
| Seasonal lakes | 0.02 | 0.00 | 0.450 | 0.02 | 0.01 | 0.764 |
| Other wetlands | 0.01 | 0.00 | 0.489 | 0.01 | 0.01 | 0.558 |
| Saline, brackish, or alkaline | 0.01 | 0.00 | 0.526 | 0.01 | 0.00 | 0.143 |

Proportions and p values for differences as tested through two-sided chi-squared tests are given for **a)** inclusion of Critically Endangered (Possibly Extinct) and Critically Endangered (Possibly Extinct in the Wild) species as extant and threatened (because their extinction has not been confirmed), and **b)** inclusion of Critically Endangered (Possibly Extinct) and Critically Endangered (Possibly Extinct in the Wild) species as extinct (because this is expected to be their true status). Otherwise, threatened species include those assessed as Critically Endangered, Endangered, or Vulnerable. Extinct freshwater species include those assessed as Extinct or Extinct in the Wild. p values marked * indicate the habitat was used by more extinct species than expected based on threatened species. p values marked † indicate the habitat was used by fewer extinct species than expected based on threatened species. Degrees of freedom = 1.

# Reporting Summary

## Statistics

For all statistical analyses, confirm that the following items are present in the figure legend, table legend, main text, or Methods section.

| n/a | Confirmed | |
|---|---|---|
| ☐ | ☒ | The exact sample size (*n*) for each experimental group/condition, given as a discrete number and unit of measurement |
| ☒ | ☐ | A statement on whether measurements were taken from distinct samples or whether the same sample was measured repeatedly |
| ☐ | ☒ | The statistical test(s) used AND whether they are one- or two-sided *Only common tests should be described solely by name; describe more complex techniques in the Methods section.* |
| ☒ | ☐ | A description of all covariates tested |
| ☒ | ☐ | A description of any assumptions or corrections, such as tests of normality and adjustment for multiple comparisons |
| ☐ | ☒ | A full description of the statistical parameters including central tendency (e.g. means) or other basic estimates (e.g. regression coefficient) AND variation (e.g. standard deviation) or associated estimates of uncertainty (e.g. confidence intervals) |
| ☐ | ☒ | For null hypothesis testing, the test statistic (e.g. *F*, *t*, *r*) with confidence intervals, effect sizes, degrees of freedom and *P* value noted *Give P values as exact values whenever suitable.* |
| ☒ | ☐ | For Bayesian analysis, information on the choice of priors and Markov chain Monte Carlo settings |
| ☒ | ☐ | For hierarchical and complex designs, identification of the appropriate level for tests and full reporting of outcomes |
| ☒ | ☐ | Estimates of effect sizes (e.g. Cohen's *d*, Pearson's *r*), indicating how they were calculated |

*Our web collection on statistics for biologists contains articles on many of the points above.*

## Software and code

Policy information about availability of computer code

| Data collection | No code was used for the data collection in this study. |
|---|---|
| Data analysis | We used the following open source software to perform the surrogacy analyses: R (v4.3), Zonation (v4, which includes the CAZ and ABF algorithms) and zonator (v0.6.0). Custom R scripts were developed to work with the software Zonation and are available at: https://zenodo.org/doi/10.5281/zenodo.10286099.<br>We also used Microsoft Excel to analyse tabular data on extinction risk, habitats, and threats. Chi-squared tests were also performed in Microsoft Excel (Version 2408). |

For manuscripts utilizing custom algorithms or software that are central to the research but not yet described in published literature, software must be made available to editors and reviewers. We strongly encourage code deposition in a community repository (e.g. GitHub). See the Nature Portfolio guidelines for submitting code & software for further information.

## Data

Policy information about availability of data

All manuscripts must include a data availability statement. This statement should provide the following information, where applicable:
- Accession codes, unique identifiers, or web links for publicly available datasets
- A description of any restrictions on data availability
- For clinical datasets or third party data, please ensure that the statement adheres to our policy

Taxonomic data for freshwater fishes are from Eschmeyer's Catalog of Fishes (http://researcharchive.calacademy.org/research/ichthyology/catalog/fishcatmain.asp) and for odonates are from World Odonata List (https://www.pugetsound.edu/slater-museum-natural-history-0/biodiversity-resources/insects/dragonflies/world-odonata-list).
All IUCN Red List assessment data are publicly available on The IUCN Red List of Threatened Species website (www.iucnredlist.org). Occasionally, where a species may be under threat from over-collection, sensitive spatial data are not publicly available. All tabular and spatial data from the IUCN Red List will be summarised and made available at www.iucnredlist.org/resources/data-repository prior to publication of the manuscript.
Baseline water stress data ('Aqueduct Water Stress Projections Data') are available from the Aqueduct Water Risk Atlas here: https://www.wri.org/data/aqueduct-water-stress-projections-data. Baseline nitrogen data ('Global - Nitrate-nitrite in Surface Water') are available from the World Bank catalogue here: https://datacatalog.worldbank.org/search/dataset/0038385/Global---Nitrate-nitrite-in-Surface-Water.

## Research involving human participants, their data, or biological material

Policy information about studies with human participants or human data. See also policy information about sex, gender (identity/presentation), and sexual orientation and race, ethnicity and racism.

| | |
|---|---|
| Reporting on sex and gender | This research does not involve human participants, their data, or biological material. |
| Reporting on race, ethnicity, or other socially relevant groupings | This research does not involve human participants, their data, or biological material. |
| Population characteristics | This research does not involve human participants, their data, or biological material. |
| Recruitment | This research does not involve human participants, their data, or biological material. |
| Ethics oversight | This research does not involve human participants, their data, or biological material. |

Note that full information on the approval of the study protocol must also be provided in the manuscript.

# Field-specific reporting

Please select the one below that is the best fit for your research. If you are not sure, read the appropriate sections before making your selection.

☐ Life sciences   ☐ Behavioural & social sciences   ☒ Ecological, evolutionary & environmental sciences

For a reference copy of the document with all sections, see nature.com/documents/nr-reporting-summary-flat.pdf

# Ecological, evolutionary & environmental sciences study design

All studies must disclose on these points even when the disclosure is negative.

| | |
|---|---|
| Study description | This study examines the extinction risk, distribution, key habitats, and drivers of threat of freshwater species (decapod crustaceans, fishes, and odonates) globally, in comparison to the status of tetrapods (amphibians, birds, mammals, and reptiles) based on data from The IUCN Red List of Threatened Species. Additionally, the study includes an analysis of the effectiveness of threatened tetrapods and an abiotic factor (water stress) as surrogates for the conservation of threatened freshwater species. |
| Research sample | The research sample was assessments of species for The IUCN Red List of Threatened Species (www.iucnredlist.org). For each species, these assessments include information on taxonomy, distribution (include a range map), population, habitats and ecology, use and trade, threats, conservation and research actions, and a Red List category of extinction risk (assigned based on Red List criteria). Data were from the 2022-2 version of the IUCN Red List, downloaded in March 2023, with additional unpublished data for a subset of freshwater fishes.<br><br>Additional datasets used were Eschmeyer's Catalog of Fishes (http://researcharchive.calacademy.org/research/ichthyology/catalog/fishcatmain.asp) and World Odonata List (https://www.pugetsound.edu/slater-museum-natural-history-0/biodiversity-resources/insects/dragonflies/world-odonata-list).<br><br>See the Methods for further details. |
| Sampling strategy | We aimed to include data on all formally described species of freshwater decapod crustaceans, fishes, and odonates. However, |

| Sampling strategy | primarily due to new descriptions since regional assessment efforts took place, the study included data on over 80% of all described freshwater species combined in the taxonomic groups of interest. This value is considered "comprehensively assessed" on the IUCN Red List and sufficient to represent the status of a taxonomic group. See the Methods for further details. |
|---|---|
| Data collection | Data were collected in the IUCN Species Information Service database (https://sis.iucnsis.org/apps/org.iucn.sis.server/SIS/index.html). Initial data compilation was primarily by species experts working remotely. This was then followed by workshops (primarily in person) where groups of species experts and facilitators with expertise in the IUCN Red List Categories and Criteria met to review and approve the data. Assessors and reviewers are listed in the credits of each assessment. The final assessments were then re-reviewed and approved by staff in the IUCN Red List Unit. See the Methods and Supplementary Information for further details. |
| Timing and spatial scale | Data collection took place between 2003 and 2023 through a series of regional assessment efforts. The frequency and duration of each assessment effort was dependent on funding available. Full details of timelines associated with each regional assessment effort can be found in the Supplementary Information. The data are global in scale, covering all areas where freshwater decapod crustaceans, fishes, and odonates are known to occur. |
| Data exclusions | No data were intentionally excluded from the analysis. However, some species were not included if, for example, they were formally described after their native region was assessed. |
| Reproducibility | Each assessment underwent two reviews. First, at least one independent scientist familiar with each species reviewed the assessment to ensure the data presented were correct and complete, and that the Red List Criteria had been applied appropriately. Once each assessment had passed this first stage of the review, including revision (if necessary), staff from the IUCN Red List Unit reviewed the assessments to ensure that the Red List Criteria had been applied appropriately, and the documentation standards had been met. Once each assessment had passed this second stage of the review, again including revision (if necessary), they were considered finalised and set for publication on the IUCN Red List website. A subset of the freshwater fish species used in this analysis had undergone only the first step of the review process described above at the time of writing. No attempts have been made to reproduce the original assessments. |
| Randomization | No randomised groups were used in the study as we aimed to cover all formally described freshwater decapod crustaceans, fishes, and odonates at the time of assessment. There were no relevant covariates to consider in the analysis. |
| Blinding | Blinding was not relevant to this study, which did not involve experimental analysis. |

Did the study involve field work?  ☐ Yes  ☒ No

# Reporting for specific materials, systems and methods

We require information from authors about some types of materials, experimental systems and methods used in many studies. Here, indicate whether each material, system or method listed is relevant to your study. If you are not sure if a list item applies to your research, read the appropriate section before selecting a response.

## Materials & experimental systems

| n/a | Involved in the study |
|---|---|
| ☒ | ☐ Antibodies |
| ☒ | ☐ Eukaryotic cell lines |
| ☒ | ☐ Palaeontology and archaeology |
| ☒ | ☐ Animals and other organisms |
| ☒ | ☐ Clinical data |
| ☒ | ☐ Dual use research of concern |
| ☒ | ☐ Plants |

## Methods

| n/a | Involved in the study |
|---|---|
| ☒ | ☐ ChIP-seq |
| ☒ | ☐ Flow cytometry |
| ☒ | ☐ MRI-based neuroimaging |

## Plants

| Seed stocks | This research does not involve plants. |
|---|---|
| Novel plant genotypes | This research does not involve plants. |
| Authentication | This research does not involve plants. |

