## [Peer Review file · Nature]

One quarter of freshwater fauna threatened with extinction

Corresponding Author: Ms Catherine Sayer

Version 0:

Reviewer comments:

Referee #1

(Remarks to the Author)

This paper fills an important gap in our understanding of state of nature – how high is the risk of extinction of species in freshwater ecosystems? It analyses new extinction risk assessments from the IUCN Red List for 23,496 species of freshwater fishes and invertebrates (decapods and dragonflies), showing that a quarter are threatened with extinction. It identifies the main threats to these species, and then shows that threatened freshwater tetrapods are reasonable surrogates for threatened freshwater species, and much better than water stress. This is relevant for conservation planning and prioritisation. The dataset will be an important addition to existing comprehensively assessed groups (mostly vertebrates) and is likely to be widely cited. The Red List methods are well established, so the novelty in this paper arises from combining the results across freshwater species, and exploring the question of surrogacy. The paper is generally clearly written and appropriately cites the literature.

My main comments are that (a) more justification is needed for including water stress, (b) comparisons should be with freshwater tetrapods rather than all tetrapods, (c) exclude Extinct in the Wild species from 'threatened' to be consistent with IUCN definitions; and (d) include Possibly Extinct and Possibly Extinct in the Wild in the analyses relating to Extinct species.

Specific comments:

Line 29: there are over 5,000 amphibians coded for inland waters, 2,000 birds and 500 mammals. You could argue the first two are 'speciose', so it might be worth qualifying the statement that 'assessments of extinction risk have been unavailable for any speciose freshwater groups'

Line 60: this sentence is about freshwaters, but I think you mean freshwater species "can be used as bioindicators of wetland quality", so some rephrasing is needed here

Line 95: does this refer to all threatened tetrapods, or the freshwater subset of these?

Line 95: are there other hydrological variables/abiotic factors that were used as surrogates for freshwater biodiversity in previous/current assessments? If so, why just focus on this one? Can any others be tested? If not, I think it would be helpful to provide a little more justification for using just this variable. It is described as "widely used", but for what? Have people really expected it to predict freshwater biodiversity patterns?

Line 104: worth noting here that older assessments may have introduced some bias and direct readers to where you explore this in line 791

Line 109: there are rather few Extinct in the Wild species, and these are not included in 'threatened' as officially defined by IUCN, so remove them to make your comparison with tetrapods more valid and in line with the literature.

Line 116: as an aside, these results provide encouraging support for the sampled approach in providing accurate estimates of extinction risk for speciose groups

Line 121: presumably the predominance of extinctions in the US is partly or mainly a reporting bias – many more must have gone extinct in other parts of the world, but were not documented, as you note in the following sentence, so you could add a note to this effect

Line 139: again, shouldn't the comparison be with threatened freshwater tetrapods? This won't change the point you are making.

Line 150: presumably lots of species are impacted by more than one threat – it might be worth pointing out that X% are impacted by multiple threats

Line 152: do you mean freshwater tetrapods?

Line 154: why do you think odonates are less susceptible to pollution, given their larvae live in water as do decapods and fishes?

Line 171: shouldn't you include PE and PEW in the analysis of drivers here as well, given these are likely to be extinct?

Line 185: Is the impact underestimated because these taxa haven't been subject to the kinds of modelling studies that have been applied to many tetrapods? Or because these groups are rather short-lived, and most declines projected under climate change are too slow to trigger Red List criteria?

Line 191: freshwater tetrapods I assume, but given you appear to have made some comparisons in the paper with all tetrapods and some with the freshwater subset, this needs clarifying here and throughout. I'd suggest using freshwater tetrapods throughout, but if there is a good reason not to, then this should be justified.

Line 202: mention why karst habitats contain so many threatened species – a combination of tiny distributions and exploitation for resources?

Line 204: why do you think this is? Presumably springs/oases because these are so small? But why lakes?

Line 221: add a sentence here to explain how the complementarity analysis was done, or what exactly 'prioritise' means in this context, referring the reader to the methods for more details

Line 222: delete "to", or maybe there are some words missing?

Line 247: there's not that much difference between 0.42 and 0.35, so maybe say "relatively poor surrogates"

Line 286: "Population pressures" presumably means the pressure from human populations, but given you've just been referring to population data (for freshwater species), so maybe rephrase

Line 313: I think there needs to be more justification in the Introduction for comparing water stress (as noted above) and more discussion here on the result that it isn't a good surrogate. Where has it been used for this purpose, and what are the implications of your finding for such uses?

Line 318: it would be worth referencing the Kunming-Montreal Global Biodiversity Framework here

Line 322-329: this is all rather generic. Consider replacing with some more specific recommendations for a couple of specific use-cases from different end-users.

Line 360: How distinctly different are the distribution of these other freshwater groups? And therefore, how valid are the study's conclusions? Some comment on this would be useful.

Line 371: this is left hanging – presumably the authors would argue that expanding this capacity is therefore important?

Fig 1 – see comments above re all vs freshwater tetrapods, and relabel the figure so it is clear that the second column refers to freshwater decapods, fishes and odonates. Is there any basis to the sequence for the latter?

Fig 2 – name the 3 freshwater groups in the figure title. What is the sequence of threats? Order from most to least prevalent for freshwater species combined? Include CR(PE) and CR(PEW) in extinct (here and in Fig 3). The numbers are not actually samples, but totals, so just say "Number of species" (here and in Fig 3).

Fig 3 – combine the classes with trivial totals into an "other" class?

Fig 4 – title is a bit clumsy. Try "Richness of threatened freshwater species". Explain here which presence, origin or season codes we included or excluded? I see this is covered in the methods, but it would be worth summarising here.

Line 617 – what does 'primary' mean here?

Line 628 – insert 'defined below' after 'preliminary assessments'? Run a couple of sensitivity tests to show they are not introducing bias. Line 702 refers to December 2023, which has now passed, so have all these now been published, and what proportion changed category before publication?

Line 631 – 21% missing is still quite substantial. Are there any data on their distribution, size, taxonomy or other characteristics to see if they are a biased subset, so you can at least acknowledge any potential biases in the results. Ditto the missing 19% of decapods. I see there is some discussion of missing species further down, so refer to this here.

Line 689 – cite the Red List Guidelines here and give the definitions of PE and PEW.

Line 781 – give the proportions for each group and overall

Line 807 Cite the Red List Guidelines or previous papers to justify using this approach

Line 825 – it would be worth recommending future reassessments apply the coding of importance so that you can focus on the most important threats and exclude trivial ones

Line 929 -what does 'freshwater quantity pressures' mean?

Referee #2

(Remarks to the Author)

Comments on Sayer et al. "Under water, under threat: extinction risk of the world's freshwater fauna"

Sayer and colleagues have produced what they describe as the first multi-taxon global freshwater biodiversity extinction risk assessment. To my knowledge, this is an accurate description. They summarize the risk and primary drivers of extinction for freshwater decapod crustaceans, fishes, and odonates (dragonflies and damselflies) on a global scale. They find that the group with the highest proportion of threatened species is the decapods (30%), followed by fishes (26%) and odonates (16%). Thus, their study provides estimates that can be widely cited by scientists and policy makers. They also assess whether a regional environmental variable that reflects human impacts (water stress) can be linked to extinction risk and found it to be a poor surrogate; they conclude rather eloquently that "setting targets around non-living nature will not be sufficient to protect and conserve living nature".

Only a year ago, the United Nations Biodiversity Conference (COP15) in Montreal recognized "inland waters" (lakes, rivers and wetlands) for the first time as a distinct realm deserving its own conservation targets. Previously, freshwater ecosystems were subsumed into the terrestrial realm based on the assumption that meeting terrestrial conservation targets would be

sufficient to protect freshwater biodiversity; evidence shows that this is not necessarily the case. In a new era of setting conservation targets it is essential to assess the global status of ubiquitous freshwater animals, as this study attempts to do. Given the recent burgeoning policy attention on what is arguably the world's most threatened (and heretofore largely ignored) biological realm, this is a very timely paper. I expect it will attract attention from a broad swath of environmental scientists, policy makers and the media.

My only substantive criticism is the unexplained omission of freshwater molluscs (gastropods and bivalves) from the study. These taxa have been assessed in multiple biogeographic regions (detailed assessments of mussels have been made in Europe, North America, and a few other biogeographic regions; e.g., Lopes-Lima et al 2018 *Hydrobiologia* 810: 1-14). I would encourage the authors to include them in their comprehensive assessment. Otherwise, their omission has undoubtedly resulted in an underestimation of freshwater diversity at risk; for example, the literature suggests a remarkable ~40% (probably conservative) of the world's known mussel taxa are imperiled.

Specific comments:

2. Lines 146-150: These proportions (totaling >100%) reflect the fact that most extinctions are the result of more than one driver. Can data be provided to show the frequency by which groups of multiple, potentially interacting stressors are implicated as drivers? The frequency by which pairs or groups of interacting stressors are deemed responsible for extinction would shed more light on the drivers of extinction and how they vary across taxa and regions. Such information cannot be inferred simply from inspection of the list of the ranked stressors.

3. Lines 234-242: It would be useful to note the scale at which tetrapods are useful surrogates for freshwater species.

4. Lines 236 & 247: It is surprising that amphibians, whose life cycles depend on freshwater, would be less representative surrogates for freshwater extinction risk than reptiles. As explained by the authors, the increased reliance of decapods and especially odonates on surrounding terrestrial habitats could contribute to increased tetrapod surrogacy. But I wonder how strong tetrapod surrogacy would be if freshwater molluscs were included in this analysis, given that mussels are longer lived and arguably more sensitive to water quality alteration than the other taxa assessed here.

5. Lines 277-292: Consider adding a stronger concluding sentence on the importance of increased monitoring of freshwater communities based on the high proportion of data deficient species—including overlooked taxa could be at high risk of extinction owing in part to their isolation.

6. Lines 306-307, Re: "meeting the needs of tetrapods cannot be assumed as sufficient to conserve freshwater species at local scales". I feel that this point should be emphasized in the bold first paragraph. It is an argument that supports recognition of inland waters as a distinct realm with respect to conservation and policy, as recognized in the Kunming-Montreal Global Biodiversity Framework.

7. Lines 359-361, Re: "it is essential that extinction risk assessments of freshwater species expand taxonomic coverage to more fully represent the realm's biodiversity..." Would the authors consider odonates to be representative of other common aquatic insect groups, such as caddisflies, stoneflies, and mayflies (groups that are typically used in biomonitoring to infer changes in water quality and ecosystem health)? I understand that conservation data may be lacking for these groups; if so, then it would be useful for the authors to identify these as major gaps in assessment.

8. Figures 2 & 3: These figures convey important information but the bars are very densely packed, making it difficult to discern patterns. Consider alternative ways of presenting the data: e.g. a six-panel figure showing the proportions for each taxa group; and a two-panel figure showing tetrapods versus freshwater taxa.

Referee #3

(Remarks to the Author)

See attachment

Version 1:

Reviewer comments:

Referee #1

(Remarks to the Author)

The authors have done a good job of responding to my previous suggestions and the comments of the other reviewers. I have only few minor wording suggestions for this revised version.

Line 104. You could clarify that these comprehensive assessments have been available for 36, 20 and 28 years respectively – rather longer than the 15 mentioned.

Line 467. This might read better as "...factors are effective surrogates". The current wording could be interpreted as being about how often they are used as surrogates.

Line 507. I'm not sure you really mean "tend to be isolated from tetrapods" i.e. occur in places with no tetrapods. Maybe

rephrase as "...tend to be in locations that differ from those for tetrapods with the smallest ranges".

Line 540. This would be better worded as "with 89 confirmed and an additional 187 suspected extinctions since 1500".

Line 544. It would be clearer to say "some threats are found to be more prevalent for freshwater species". "Greater" could be taken to relate to the magnitude of the impact of each threat to each species.

Line 638. It would be clearer to say "are very poor surrogates when used in conservation planning for threatened freshwater species" – I don't think you mean that they are surrogates for conservation per se.

Line 639. Insert "The distribution of" before "biodiversity"?

Line 654. Readers may not be familiar with this tool. So perhaps reword as "For example, private sector users of the Integrated Biodiversity Assessment Tool (IBAT) can run a 'freshwater report' for areas of interest which uses IUCN Red List data..."

Line 1537. Add some text here on the logic behind Extended Data Table 1 etc, e.g. "We calculated the proportion of species impacted by each threat type for extant and extinct species in two ways: firstly treating CR(PE) and CR(PEW) species as extant (because their extinction has not been confirmed), and secondly treating them as extinct (because this is suspected to be their true status)" – or something along these lines.

Referee #3

(Remarks to the Author)

The authors have done a commendable job addressing my primary comments, particularly around improving the clarity of representation of the study's methods and potential limitations which would otherwise be limited by the format used in this journal. I am satisfied with the revisions conducted in response to my prior comments and consider the manuscript greatly improved. I have no further comments and recommend acceptance at this stage.

Responses to referees' comments

Referee #1 (Remarks to the Author):

This paper fills an important gap in our understanding of state of nature – how high is the risk of extinction of species in freshwater ecosystems? It analyses new extinction risk assessments from the IUCN Red List for 23,496 species of freshwater fishes and invertebrates (decapods and dragonflies), showing that a quarter are threatened with extinction. It identifies the main threats to these species, and then shows that threatened freshwater tetrapods are reasonable surrogates for threatened freshwater species, and much better than water stress. This is relevant for conservation planning and prioritisation. The dataset will be an important addition to existing comprehensively assessed groups (mostly vertebrates) and is likely to be widely cited. The Red List methods are well established, so the novelty in this paper arises from combining the results across freshwater species, and exploring the question of surrogacy. The paper is generally clearly written and appropriately cites the literature.

My main comments are that (a) more justification is needed for including water stress, (b) comparisons should be with freshwater tetrapods rather than all tetrapods, (c) exclude Extinct in the Wild species from ‘threatened’ to be consistent with IUCN definitions; and (d) include Possibly Extinct and Possibly Extinct in the Wild in the analyses relating to Extinct species.

Specific comments:

Line 29: there are over 5,000 amphibians coded for inland waters, 2,000 birds and 500 mammals. You could argue the first two are ‘speciose’, so it might be worth qualifying the statement that ‘assessments of extinction risk have been unavailable for any speciose freshwater groups’

Response: We have reworded this statement to make it clear the focus is on taxonomic groups for which the majority of species primarily live within freshwaters.

Line 60: this sentence is about freshwaters, but I think you mean freshwater species “can be used as bioindicators of wetland quality”, so some rephrasing is needed here.

Response: Your understanding is correct. We have rephrased the text accordingly.

Line 95: does this refer to all threatened tetrapods, or the freshwater subset of these?

Response: All analyses in this study are comparing freshwater species (defined as freshwater decapod crustaceans, fishes, and odonates) to all tetrapods (defined as amphibians, birds, mammals, and reptiles; regardless of their realm). One aim of this study is to see if the data that are currently being used to make many decisions related to biodiversity conservation are appropriate to represent the needs of freshwater fauna. When looking at species data, it is currently data on tetrapods (regardless of realm) that are being used to inform decisions, and not data on freshwater tetrapods being used to inform decisions on other freshwater species. This is why we have chosen these two species groups for comparison, noting also that, while around a quarter of tetrapods are coded in the Red List as using freshwaters, they are a predominantly terrestrial group. We have aimed to make this clearer in the text, including by replacing some instances of “freshwater species” with “freshwater decapods, fishes, and odonates” to make it clearer that the comparison is primarily taxonomic. The former is also defined in the Introduction.

Line 95: are there other hydrological variables/abiotic factors that were used as surrogates for freshwater biodiversity in previous/current assessments? If so, why just focus on this one? Can any others be tested? If not, I think it would be helpful to provide a little more justification for using just this variable. It is described as “widely used”, but for what? Have people really expected it to predict freshwater biodiversity patterns?

Response: Water stress is very widely used as a surrogate in freshwater biodiversity conservation. In addition to use by the Science Based Targets Network (originally referenced), it is also used as SDG indicator 6.4.2 and by GRI, TNFD, WWF Water Risk Filter, and Global Compact. All of these additional uses are now referenced in the text. To complement water stress, we have also added a surrogacy analysis using an abiotic factor to represent water quality. We have used a global nitrogen layer, representing eutrophication levels, in this analysis and the results support that of the original analysis using water stress.

Line 104: worth noting here that older assessments may have introduced some bias and direct readers to where you explore this in line 791

Response: We have added text to direct readers to the Methods for discussions of potential biases introduced by this timespan.

Line 109: there are rather few Extinct in the Wild species, and these are not included in ‘threatened’ as officially defined by IUCN, so remove them to make your comparison with tetrapods more valid and in line with the literature.

Response: Threatened species are those assessed as Vulnerable, Endangered, or Critically Endangered. This is written in the text. However, for calculation of the best estimate of percentage threatened, Annex I of the Guidelines for Appropriate Uses of IUCN Red List Data (https://nc.iucnredlist.org/redlist/content/attachment_files/Guidelines_for_Reporting_Proportion_Threatened_ver_1_2.pdf) states that Extinct in the Wild species should be included because they would be downlisted to a threatened category if successfully introduced. It is therefore correct, and in keeping with IUCN guidance, to include Extinct in the Wild species in this calculation, even though Extinct in the Wild is not technically a threatened category. The calculations for the freshwater species groups and tetrapods have used the same formula and therefore are comparable. This method has been followed in other recent IUCN Red List analyses (e.g. Luedtke et al. 2023)

Line 116: as an aside, these results provide encouraging support for the sampled approach in providing accurate estimates of extinction risk for speciose groups

Response: Agreed. On this basis, we have added text on the results of the sampled Red List approach for freshwater molluscs, which indicates close to a third of species are threatened. We could not include this group in our analysis because only c. 50% of species are currently assessed globally, with notable gaps in their geographic representation, meaning their incorporation would introduce regional and taxonomic biases.

Line 121: presumably the predominance of extinctions in the US is partly or mainly a reporting bias – many more must have gone extinct in other parts of the world, but were not documented, as you note in the following sentence, so you could add a note to this effect

Response: We have added a note to indicate this could be a reporting bias, but also acknowledging that other regions with similarly high data availability (e.g., Europe) do not show such high numbers of extinctions).

Line 139: again, shouldn’t the comparison be with threatened freshwater tetrapods? This won’t change the point you are making.

Response: Please see the response above on line 95.

Line 150: presumably lots of species are impacted by more than one threat – it might be worth pointing out that X% are impacted by multiple threats

Response: 84% of threatened species are impacted by more than one threat. We have added this to the text.

Line 152: do you mean freshwater tetrapods?

Response: Please see the response above on line 95.

Line 154: why do you think odonates are less susceptible to pollution, given their larvae live in water as do decapods and fishes?

Response: Pollution is primarily a threat to odonates in their larval stages, whereas the other freshwater groups are impacted by it throughout their life cycles. Habitat loss impacts odonates throughout their life cycles. We have expanded the text on these threats.

Line 171: shouldn't you include PE and PEW in the analysis of drivers here as well, given these are likely to be extinct?

Response: To avoid double counting these species in the analyses, we need to consider them either extant and threatened, or extinct. Possibly Extinct and Possibly Extinct in the Wild species are normally considered extant, primarily to avoid the Romeo Error (Collar 1998) whereby any conservation measures are removed from threatened species on the mistaken belief that the species have already gone extinct. This is the reason these tags were initially developed (Butchart et al. 2006). We have therefore retained Possibly Extinct and Possibly Extinct in the Wild species as extant species throughout the main text; this is also in keeping with the fact that Possibly Extinct and Possibly Extinct in the Wild are tags in the Critically Endangered category. However, we have additionally redone the threats and habitats analyses and the richness maps, instead considering these species as extinct and incorporated the results in the Extended Data tables and figures. Any differences between the findings are then discussed in the main text.

Line 185: Is the impact underestimated because these taxa haven't been subject to the kinds of modelling studies that have been applied to many tetrapods? Or because these groups are rather short-lived, and most declines projected under climate change are too slow to trigger Red List criteria?

Response: This is due to the lack of modelling studies on the impacts of climate change on freshwater species. We have clarified this in the text and added an extra supporting reference (Mancini et al. 2024).

Line 191: freshwater tetrapods I assume, but given you appear to have made some comparisons in the paper with all tetrapods and some with the freshwater subset, this needs clarifying here and throughout. I'd suggest using freshwater tetrapods throughout, but if there is a good reason not to, then this should be justified.

Response: Please see the response above on line 95.

Line 202: mention why karst habitats contain so many threatened species – a combination of tiny distributions and exploitation for resources?

Response: Yes, this is the result of both many small populations of restricted range species and a variety of threats leading to degradation of the habitat. We have added this to the text.

Line 204: why do you think this is? Presumably springs/oases because these are so small? But why lakes?

Response: Yes, springs and oases are generally restricted in range. Both these habitats and lakes often have high incidence of threatening activities, such as invasive species, water extraction, and

overharvesting, from which species endemic to them then cannot escape. We have added this to the text.

Line 221: add a sentence here to explain how the complementarity analysis was done, or what exactly ‘prioritise’ means in this context, referring the reader to the methods for more details

Response: We have expanded this text and added a reference to the Methods section.

Line 222: delete “to”, or maybe there are some words missing?

Response: We have deleted “to”.

Line 247: there’s not that much difference between 0.42 and 0.35, so maybe say “relatively poor surrogates”

Response: We have rewritten this sentence. The descriptors used are based on ranges of values that were defined in the Methods section but have now also been added to the main text for clarity.

Line 286: “Population pressures” presumably means the pressure from human populations, but given you’ve just been referring to population data (for freshwater species), so maybe rephrase

Response: It is correct that this refers to pressures from human populations. We have rephrased this for clarity.

Line 313: I think there needs to be more justification in the Introduction for comparing water stress (as noted above) and more discussion here on the result that it isn’t a good surrogate. Where has it been used for this purpose, and what are the implications of your finding for such uses?

Response: Please see the response above on line 95 for justification on use of water stress and its current uses. We have added more detail to the Discussion on the implications of this poor surrogacy, noting the need to re-evaluate existing conservation strategies that rely on these abiotic factors.

Line 318: it would be worth referencing the Kunming-Montreal Global Biodiversity Framework here

Response: We have added this reference.

Line 322-329: this is all rather generic. Consider replacing with some more specific recommendations for a couple of specific use-cases from different end-users.

Response: We have added two examples of how the freshwater species data can be interrogated through the Integrated Biodiversity Assessment Tool.

Line 360: How distinctly different are the distribution of these other freshwater groups? And therefore, how valid are the study’s conclusions? Some comment on this would be useful.

Response: We have removed this statement because the scale of analysis in the cited reports is different than that analysed here. However, we have added new reasoning to support the inclusion of these additional taxonomic groups.

Line 371: this is left hanging – presumably the authors would argue that expanding this capacity is therefore important?

Response: Yes, we have added an extra sentence with this recommendation.

Fig 1 – see comments above re all vs freshwater tetrapods, and relabel the figure so it is clear that the second column refers to freshwater decapods, fishes and odonates. Is there any basis to the sequence for the latter?

Response: Please see the response above on line 95 regarding all versus freshwater tetrapods. We have added an additional label to indicate the latter three columns refer to freshwater species. These three groups are in alphabetical order.

Fig 2 – name the 3 freshwater groups in the figure title. What is the sequence of threats? Order from most to least prevalent for freshwater species combined? Include CR(PE) and CR(PEW) in extinct (here and in Fig 3). The numbers are not actually samples, but totals, so just say “Number of species” (here and in Fig 3).

Response: Figures 2 and 3 are now Tables 1 and 2. We have added the three freshwater groups in the title and replaced “Sample sizes” with “Number of species”. The threats and habitats have been ordered by most to least prevalent for freshwater species combined. Please see the response above on line 171 regarding CR (PE) and CR (PEW) species.

Fig 3 – combine the classes with trivial totals into an “other” class?

Response: We have combined the classes with trivial totals into “Other wetlands”

Fig 4 – title is a bit clumsy. Try “Richness of threatened freshwater species”. Explain here which presence, origin or season codes we included or excluded? I see this is covered in the methods, but it would be worth summarising here.

Response: We have updated the title and added the presence, origin, and seasonality codes used to all richness maps.

Line 617 – what does ‘primary’ mean here?

Response: These are decapods that spend their entire lifecycle in freshwaters. We have added this to the text.

Line 628 – insert ‘defined below’ after ‘preliminary assessments’? Run a couple of sensitivity tests to show they are not introducing bias. Line 702 refers to December 2023, which has now passed, so have all these now been published, and what proportion changed category before publication?

Response: Eighty-seven per cent (1,370 species) of these preliminary assessments are now published, with only seven species (0.5%) changing Red List Category prior to publication as a result of the second stage of review. We expect the remaining 207 species (13%) with preliminary assessments to have completed the assessment process by October 2024. We have added this information in the ‘Red List assessment process’ section, and added “defined below” after the first mention of preliminary assessments.

Line 631 – 21% missing is still quite substantial. Are there any data on their distribution, size, taxonomy or other characteristics to see if they are a biased subset, so you can at least acknowledge any potential biases in the results. Ditto the missing 19% of decapods. I see there is some discussion of missing species further down, so refer to this here.

Response: We have added text indicating the subsequent section that describes the characteristics of the missing species.

Line 689 – cite the Red List Guidelines here and give the definitions of PE and PEW.

Response: We have added the definitions and the reference to the Red List Guidelines.

Line 781 – give the proportions for each group and overall

Response: We have moved text on spatial data availability from later in the Methods section to this subsection.

Line 807 Cite the Red List Guidelines or previous papers to justify using this approach

Response: We have expanded the text to give more background on the reason to report a range of values for the percentage of threatened species, with reference to the Guidelines on appropriate use of IUCN Red List data.

Line 825 – it would be worth recommending future reassessments apply the coding of importance so that you can focus on the most important threats and exclude trivial ones

Response: We have added this recommendation.

Line 929 -what does ‘freshwater quantity pressures’ mean?

Response: We have replaced this with “water demand compared to available renewable water and groundwater”.

Responses to referees' comments

Referee #2 (Remarks to the Author):

Comments on Sayer et al. "Under water, under threat: extinction risk of the world's freshwater fauna"

Sayer and colleagues have produced what they describe as the first multi-taxon global freshwater biodiversity extinction risk assessment. To my knowledge, this is an accurate description. They summarize the risk and primary drivers of extinction for freshwater decapod crustaceans, fishes, and odonates (dragonflies and damselflies) on a global scale. They find that the group with the highest proportion of threatened species is the decapods (30%), followed by fishes (26%) and odonates (16%). Thus, their study provides estimates that can be widely cited by scientists and policy makers. They also assess whether a regional environmental variable that reflects human impacts (water stress) can be linked to extinction risk and found it to be a poor surrogate; they conclude rather eloquently that "setting targets around non-living nature will not be sufficient to protect and conserve living nature".

Only a year ago, the United Nations Biodiversity Conference (COP15) in Montreal recognized "inland waters" (lakes, rivers and wetlands) for the first time as a distinct realm deserving its own conservation targets. Previously, freshwater ecosystems were subsumed into the terrestrial realm based on the assumption that meeting terrestrial conservation targets would be sufficient to protect freshwater biodiversity; evidence shows that this is not necessarily the case. In a new era of setting conservation targets it is essential to assess the global status of ubiquitous freshwater animals, as this study attempts to do. Given the recent burgeoning policy attention on what is arguably the world's most threatened (and heretofore largely ignored) biological realm, this is a very timely paper. I expect it will attract attention from a broad swath of environmental scientists, policy makers and the media.

My only substantive criticism is the unexplained omission of freshwater molluscs (gastropods and bivalves) from the study. These taxa have been assessed in multiple biogeographic regions (detailed assessments of mussels have been made in Europe, North America, and a few other biogeographic regions; e.g., Lopes-Lima et al 2018 *Hydrobiologia* 810: 1-14). I would encourage the authors to include them in their comprehensive assessment. Otherwise, their omission has undoubtedly resulted in an underestimation of freshwater diversity at risk; for example, the literature suggests a remarkable ~40% (probably conservative) of the world's known mussel taxa are imperiled.

Response: We have not included freshwater molluscs in our study because the group is not yet comprehensively assessed. At present only c. 50% of all formally described freshwater mollusc species have global assessments published on the IUCN Red List and there are large gaps in the coverage of species native to the Americas, Russia, Asia, and Oceania. Any global analysis of the group would therefore be biased (geographically and likely, taxonomically) and this use is not supported by the Guidelines for Appropriate Uses of IUCN Red List Data. Therefore, we have not added freshwater molluscs to the taxonomic groups investigated in this study. However, we have referenced both Lopes-Lima et al. 2018 and Bohm et al. 2021, which investigate the extinction risk of this group based on a sample, and noted that these point to the freshwater molluscs being more threatened than the freshwater decapods, fishes, or odonates. Completion of a global freshwater mollusc assessment is recommended in the Discussion and we hope this to be fulfilled by 2027 through an ongoing assessment project. At this time we will be able to assess the status of the group globally.

Specific comments:

2. Lines 146-150: These proportions (totaling >100%) reflect the fact that most extinctions are the result of more than one driver. Can data be provided to show the frequency by which groups of multiple, potentially interacting stressors are implicated as drivers? The frequency by which pairs or groups of interacting stressors are deemed responsible for extinction would shed more light on the

drivers of extinction and how they vary across taxa and regions. Such information cannot be inferred simply from inspection of the list of the ranked stressors.

Response: Over two-thirds of extinctions of freshwater species were recorded to be caused by more than one threat in their Red List assessments. We have added an analysis of the frequency of co-occurring pairs of threats to this section, highlighting the threats that commonly interact to drive extinctions.

3. Lines 234-242: It would be useful to note the scale at which tetrapods are useful surrogates for freshwater species.

Response: We have added the scale at which surrogacy was investigated – c. 50 x 50 km resolution.

4. Lines 236 & 247: It is surprising that amphibians, whose life cycles depend on freshwater, would be less representative surrogates for freshwater extinction risk than reptiles. As explained by the authors, the increased reliance of decapods and especially odonates on surrounding terrestrial habitats could contribute to increased tetrapod surrogacy. But I wonder how strong tetrapod surrogacy would be if freshwater molluscs were included in this analysis, given that mussels are longer lived and arguably more sensitive to water quality alteration than the other taxa assessed here.

Response: Many of the threatened amphibians have narrow distributions which likely means their distributions do not coincide with many threatened freshwater species, so reducing their efficacy as surrogates. It would be interesting to test surrogacy of tetrapods for freshwater molluscs but this is unfortunately not possible at present at a global scale, due to the fact that freshwater molluscs are not yet comprehensively assessed (see response above for further information).

5. Lines 277-292: Consider adding a stronger concluding sentence on the importance of increased monitoring of freshwater communities based on the high proportion of data deficient species—including overlooked taxa could be at high risk of extinction owing in part to their isolation.

Response: We have rewritten this paragraph and included a concluding sentence as recommended.

6. Lines 306-307, Re: “meeting the needs of tetrapods cannot be assumed as sufficient to conserve freshwater species at local scales”. I feel that this point should be emphasized in the bold first paragraph. It is an argument that supports recognition of inland waters as a distinct realm with respect to conservation and policy, as recognized in the Kunming-Montreal Global Biodiversity Framework.

Response: We have reworded the bold first paragraph to include this information as the closing remark.

7. Lines 359-361, Re: “it is essential that extinction risk assessments of freshwater species expand taxonomic coverage to more fully represent the realm’s biodiversity...” Would the authors consider odonates to be representative of other common aquatic insect groups, such as caddisflies, stoneflies, and mayflies (groups that are typically used in biomonitoring to infer changes in water quality and ecosystem health)? I understand that conservation data may be lacking for these groups; if so, then it would be useful for the authors to identify these as major gaps in assessment.

Response: We have expanded this section to include the recommendation to assess caddisflies, stoneflies, and mayflies.

8. Figures 2 & 3: These figures convey important information but the bars are very densely packed, making it difficult to discern patterns. Consider alternative ways of presenting the data: e.g. a six-panel figure showing the proportions for each taxa group; and a two-panel figure showing tetrapods versus freshwater taxa.

Response: We have reformatted these figures into tables/grids for clarity.

Responses to referees' comments

Referee #3:

Questions from the Editor:

Does the manuscript have flaws which should prohibit its publication?

The articles' objectives, results, and their implications must be more clearly and accurately described (see detailed comments below).

Do you feel that the results presented are of immediate interest to many people in your own discipline, or to people from several disciplines?

Yes, the results are of general interest to professionals in my field and other environmental fields. The loss of global biodiversity *should* be of interest to a general audience and this type of study and the associated messaging deserve much greater attention from the scientific community at large.

Overall Comments

The manuscript reflects a tremendous amount of long-term effort by more than a thousand scientists bringing together data of mixed quality and engaging in a well-established expert solicitation process. The conservation issues covered, notably the freshwater biodiversity crisis and its severity across space, are paramount for environmental and sustainability sciences. I feel that the broader objectives of the paper should be more clearly stated, especially early on. In addition, more broad/general references to the approach used and at least the conceptual basis of the methods employed should be more clear outside of the methods section. For an interdisciplinary audience reading a paper formatted with the methods section last, the authors must be realistic that most readers will not see the methods section narrative. Accordingly they should be more clear about the nature of the investigation (in particular the nature of the quantitative data used, and the degree of reliance on expert opinion) and more carefully phrase, interpret, and qualify results in the discussion session. I think substantial revisions to the text will be necessary to accomplish these goals but the paper is absolutely worthy of publication and the research is of substantial scientific merit for the broader environmental conservation community. The title is also vague uninformative as to the study's findings.

Response: We have revised the title to include specific mention of one of the main findings of the manuscript that a quarter of freshwater fauna species are threatened with extinction.

We have expanded the final paragraph of the Introduction to better explain the broader objectives of the paper and therefore, support the analyses undertaken.

We have added details to the main text of the IUCN Red List methodology, with reference to the Methods for full details and discussion of biases.

We have revisited our language in how results are presented throughout the text.

Comments on Text

Bold First Paragraph

28-29 – How can the authors reconcile this statement with recent studies on population declines in freshwater megafauna, e.g., **He et al., 2019**? Is the idea that this was merely an analysis of trends and not extinction risk per se? If so then the authors should specify early on what is meant by 'extinction risk' in this context. As currently used with little qualification (here or in the main text), it implies that the IUCN standard methodologies are the *de-facto* definition of extinction risk, which is a bit

misleading for non-experts. In the main text especially, a thorough discussion of the advantages and disadvantages/limitations of the IUCN classification system is worthwhile.

In this paragraph, at the very least, authors should specify that they are specifically referencing the IUCN definition of risk and explain that it is a semi-quantitative rating system. The authors also seem to use the term somewhat interchangeably with the umbrella term “biodiversity assessment”, and this distinction or shared definition should be clarified.

Response: We have clarified in this paragraph that we are referring to assessments of extinction risk on the IUCN Red List. We have also reworded the text to make it clear that global scale data on speciose freshwater groups have been largely missing until now, hence making it clearer why studies such as He et al. 2019 are not considered (i.e., given the scale and scope of coverage in terms of number of species, geographic coverage, and taxonomic representation).

In the main text, we have provided more information on the IUCN Red List methodology, with reference to the Methods for full details and for a discussion of potential biases.

36 – I think the word “reveals” is a bit heavy-handed here. There is no doubt that freshwater biodiversity is in crisis, but my understanding of the analyses conducted is that they *suggest* that this proportion of species are threatened with extinction. Actual quantitative assessments of population trends and demographic rates were not undertaken. The authors must find an appropriate balance between promoting the impressive, laudable scale of the research undertaken, the gravity of the findings, and the limitations of the analytical approaches. This is especially true considering that the methods are relegated to the end of the manuscript in this format and a small proportion of readers will realistically ever look through them.

Response: We have revisited our language throughout the text in light of this comment.

Main Text

79 – it is around this point that the authors should clarify specifics of the approach for IUCN risk assessment/biodiversity assessment; this should be considered and treated as important background information and separate from / less detailed than the methods. The implications, limitations, and caveats of these assessments should also be described.

It should be clear to readers that assessments are undertaken by convening panels for a synthesis of expert opinion and not purely quantitative analyses.

Response: We have added here a high-level description of the IUCN Red List methodology, with reference to the Methods for full details and for a discussion of potential biases.

129 – Although the authors briefly touch upon the point, I think it should be discussed whether the high number of documented extinctions in the United States is an artifact to better data availability.

Response: We have added a note to indicate this could be a reporting bias, but also acknowledging that other regions with similarly high data availability (e.g., Europe) do not show such high numbers of extinctions.

163-167 – This reported result is described with a much greater degree of interpretation and support than many preceding points, and cites more literature explaining details about why/how this particular threat is likely so prominent. Other described results like the preceding ones pertaining to impacts on decapods and odonates should be treated with similar attention. Alternatively if such information on the nature of these threats is not well understood, it should be clearly stated by the authors. At present there is no clear reason for this differential treatment of findings.

Response: We have rewritten this section on the threat of dams and water management to focus on the types of impacts on freshwater species (e.g. habitat loss and degradation, with some more detailed examples of how this is occurring taken from the previous text) for consistency with the text above on

odonates and decapods, plus the text below on climate change. The sections on dams and water management and climate change are longer because of the wider variety of impact types caused by the threats. All sections are supported by references.

171-179 – What can the authors say about the implications of this discrepancy? Does it imply anything about changing pressures on freshwater systems, or a filter effect on biota across time?

Response: This highlights the increased severity of impacts of some threats on freshwater populations. We have added this to the text. Extinctions on the IUCN Red List are not assigned based on a time period over which a species is not seen, but rather when exhaustive surveys have failed to find evidence of the species – this allows survey effort to be taken into account when stating a species as extinct. This definition means we cannot look at the degree of threats over time in relation to extinct species.

191 – the geographic spread/habitat specialization of these threatened odonates might be worth adding here for context. It is not intuitively clear why forest loss would be such an important driver of their declines. Is it from watershed impacts of deforestation or do they actually occupy forest habitats? Are they found in the actual forests or in wetland habitats embedded in the forests? Are wetland habitats classed as “Forest” swamps? Streams running through forests? Mangroves?

Response: A high proportion of odonate species use forests in their adult phases for shelter and as hunting grounds. Freshwater habitats are always used for reproduction and support the larval stages. These freshwater habitats may be within or outside of forest habitats. It is predominantly sub(tropical) species which rely fully on forests and freshwater habitats within them for their life cycles. In a Red List assessment, we would include both a “forest” habitat type and a “wetland” habitat type to reflect this. We have updated the text to clarify the above.

233 – the authors may want to comment on and perhaps provide a threshold on what they consider a “good” surrogate here. Or, they could provide ranges of quality for surrogates based on SAI value. For example, it sounds like anything below 0.40 is considered “poor”, while values below 0.5 and above 0.4 are considered “reasonable”, etc.

Response: We previously included ranges with descriptors in the Methods section but have now added them in the main text too for clarity.

254 – Are any of these values sufficiently negative to be useful? I.e., if the relationship is worse than random, can it be informative in the inverse? This should be clarified.

Response: We have expanded the text on this result. A negative SAI value indicates that the surrogate's performance in predicting or representing the target conservation strategy is worse than what would be expected if one were to select species or factors at random. If these abiotic factors are being used to guide conservation decisions, they may lead to incorrect conclusions and ineffective management actions. For example, managing nitrogen levels based on the assumption that it correlates with a conservation strategy might not improve, and could even harm, biodiversity. Conservation strategies that rely on water stress and nitrogen as key indicators should be re-evaluated.

Discussion

259-261 – Again, I think the authors should be clear about what mechanistically is meant by “assessing extinction risk” here. Is it quantitative? Semiquantitative? This will not be common knowledge to a broad audience. That is, the authors should be more clear about the nature of data collection and assessment in the study, and not treat it as some sort of definitive analysis. I certainly don't disagree with the importance of the findings, nor do I question the immense effort of the undertaking, but language is needed to clarify that this is largely the result of expert consultation, so findings represent a strong scientific consensus (which is nevertheless valuable).

Phrases like “Although agriculture and invasive species are major threats...” are in their phrasing overstretched. At least given the understanding conveyed by the methods (and if this is not true, then the authors should substantially clarify the methods), it would be more appropriate to describe study findings as “Although agriculture and invasive species *are widely understood [or considered] to be* major threats...” etc.

Response: *We have revisited our language throughout the text in light of this comment.*

274 – Missing a recent key citation on the potential applications of NbS to the freshwater biodiversity crisis <https://doi.org/10.1371/journal.pwat.0000126>

Response: *We have added this reference.*

268-271 – There are many papers from the last 1-3 years on this topic, and some merit acknowledgment here. The better integration of ecological research, biodiversity conservation, and water resources management has been repeatedly proposed in the Water Diplomacy / IWRM literature (ecological stakeholder analog framework). Vollmer and others have advocated for watershed health as a way of combining broader water management and environmental goals under an integrated, local-scale decision-making lens.

Response: *We have added referenced examples of a couple of approaches for integration.*

277 – This is a revealing sentence that highlights what should be clarified in the definition of “extinction risk” necessary in earlier segments of the paper. Given Nature’s format of having methods at the end of the paper, at least some brief explanation of this term, clarifying that it is not a quantitative viability analysis, is thus important early in this manuscript.

Response: *We now include early on in the main text a description of the five criteria used to assess extinction risk. This is followed up by more detail here on which are frequently applied for freshwater species and why.*

282 – Is restricted range *relative to historical baselines* considered in this context? Could it be more or less helpful in assessing freshwater species which naturally have small ranges?

Response: *The majority of threatened freshwater species are assessed as such based on criteria B and D2, which highlight species with restricted geographic ranges that are facing threats. The ranges are based on current ranges and do not take account of historical ranges (i.e., a naturally range restricted species and a species with a restricted range as a result of habitat/population loss would be treated the same by these criteria). However, another criterion (A) highlights species that have undergone, are undergoing, or will face population declines, for example due to loss of habitat. This criterion would better highlight species that are not naturally restricted range. Together these cover different symptoms of extinction risk.*

292 – citizen science and other participatory monitoring at local scales warrant mention here as a potential stopgap for the high need and low available funding for monitoring. The authors should cite some examples of what “improved regulation” should look like, as well as involvement of stakeholders. There is a growing literature of work on stakeholder involvement in freshwater resources management and biodiversity conservation.

Response: *We have reworded the text to add some more specific examples of stakeholders and added more references to support their involvement in data generation. We have added a new sentence specifically on the potential role of citizen science.*

312 etc. – Since the broader focus of this paper is on spatial representation and areal coverage, there should be some discussion of caveats, e.g., nonuniform distribution of organisms within these ranges,

disproportionate importance of certain areas/habitat types within ranges for life history / population dynamics, and so on.

Response: We have added a sentence on this to the Discussion. Non-uniform distributions are also already discussed in the mapping section of the Methods.

313 – the poor alignment of water stress and freshwater biodiversity threat status is too large a result to warrant such short discussion. Why do the authors think that this integrated measure, even if abiotic, does such a poor job? What biologically important factors is it missing?

Response: Biodiversity is influenced by a complex interplay of various biotic and abiotic factors, meaning that it will be hard to replicate through an abiotic factor or factors alone. We have added an additional abiotic factor to the analyses, which shows the same poor surrogacy. The discussion of these factors has been expanded.

327 – what is “cross-sectorial” meant to signify? Which additional sectors should be brought in, and why? There is recent literature for example on overlaps between biodiversity conservation and water resources management, agriculture, infrastructure, and so on.

Response: We have added a reference and mentioned specific sectors that should be coordinated with (e.g. agriculture and energy) given their demonstrated impacts on and reliance on freshwaters

336 – literature on relational value of biodiversity and ecosystems should be cited here. Another great reference not to be omitted is Cooke et al (2021) freshwater ethic - <https://doi.org/10.1002/aqc.3537>

Response: We have added this reference.

347 – how long is “a while”?

Response: We have reworded this sentence. The original wording implied it was data availability and timing that had allowed KBAs to be identified for freshwater species in some regions, whereas in fact it is resources (funding) for work in particular regions that allowed KBAs to be identified.

354 – The narrative of this paper seems to primarily circulate around protected areas as the major conservation action of concern. I understand if this is the M.O. of IUCN, but it is worth some discussion of other conservation strategies like traditional agriculture, NbS, invasive species removal, and so on, as the protected area paradigm is coming under heavy criticism for its social equity implications in recent years. That is also worth specific mention for non-expert readership.

Response: We do not feel that the narrative focuses on protected areas as the main conservation action. Protected areas are only mentioned once in the manuscript, in this highlighted paragraph. We also mention “other protection and management mechanisms” which would cover the other conservation actions mentioned. Other conservation actions and strategies (e.g. NbS) are also discussed elsewhere in the Discussion.

Figures

Y-axis label for Figure 1; The quantity “Species per red list category” cannot have the unit (%); I think what the authors mean is Proportion of species per red list category.

Response: We have relabelled the y axis “Proportion of species per Red List Category (%)”

Figure 2 & 3- could this information be more effectively displayed as a table or as a grid with

color-coded cells to highlight results of particular importance? This number of bars across the X axis seems unwieldy. Alternatively, perhaps authors could consider using arrows or boxes to highlight the key take-homes points within these figures.

Response: We have reformatted these figures into tables/grids for clarity.

Figure 3 – Discussion of the findings of this analysis outside of the methods section should clarify whether species occupying multiple habitat types were “double counted”. The authors should be cautious of the potential for readers to infer from these results that certain habitat types are more worthy of conservation than others simply on the basis of (threatened) species richness metrics.

Response: The main text indicates that most species occur in more than one habitat type. Patterns of habitat use were similar amongst all freshwater species and threatened freshwater species, with the exception of karst which is highlighted in the text. We have also highlighted habitats that are hotspots of extinction. Together these results can be used to highlight a range of key habitats depending on aims of conservation actions.

Methods

623 – What did these tabular and spatial data consist of? Occurrences? Counts?

Response: We have added text indicating the subsequent sections that describe the data used.

679-80 – What were the thematic classes used for each of these categories and how were they assigned?

Response: We have added text indicating the information categories follow those in the IUCN Red List supporting information guidelines, with the document referenced.

881 – some brief (1-2 sentence) explanation of the function and statistical mechanism of SAIs would be helpful here. The formula is described below, but the interpretation and utility should be discussed.

Response: We have added a few lines on the function and mechanism of SAIs, with the detailed explanation retained later in the section.

933 – This is worth mentioning in the discussion paragraph around the results of water stress. Were there indeed no freshwater species in these areas, or are they already missing due to a filter effect?

Response: We have clarified the wording here as these statements relate to threatened freshwater species, which are those considered in the surrogacy analysis. There are large parts of the world without threatened freshwater species (see Figure 2) and therefore, it is not a surprise that there were a high proportion of cells with water stress or nitrogen data, but no threatened freshwater species.

Responses to referee comments

Referee #1:

The authors have done a good job of responding to my previous suggestions and the comments of the other reviewers. I have only few minor wording suggestions for this revised version.

We thank the referee again for their original suggestions and are pleased that they are satisfied with our responses. Thank you also for the helpful wording suggestions provided here to improve clarity of the manuscript.

Line 104. You could clarify that these comprehensive assessments have been available for 36, 20 and 28 years respectively – rather longer than the 15 mentioned.

To be concise, we have updated the wording to say these comprehensive assessments have been available for over 20 years.

Line 467. This might read better as “...factors are effective surrogates”. The current wording could be interpreted as being about how often they are used as surrogates.

We have added “effective” as suggested.

Line 507. I'm not sure you really mean “tend to be isolated from tetrapods” i.e. occur in places with no tetrapods. Maybe rephrase as “...tend to be in locations that differ from those for tetrapods with the smallest ranges”.

We have rephrased this text as suggested.

Line 540. This would be better worded as “with 89 confirmed and an additional 187 suspected extinctions since 1500”.

We have rephrased this text as suggested.

Line 544. It would be clearer to say “some threats are found to be more prevalent for freshwater species”. “Greater” could be taken to relate to the magnitude of the impact of each threat to each species.

We have replaced “greater” with “more prevalent”.

Line 638. It would be clearer to say “are very poor surrogates when used in conservation planning for threatened freshwater species” – I don't think you mean that they are surrogates for conservation per se.

We have rephrased this text as suggested.

Line 639. Insert “The distribution of” before “biodiversity”?

We have added “The distribution of” as suggested.

Line 654. Readers may not be familiar with this tool. So perhaps reword as “For example, private sector users of the Integrated Biodiversity Assessment Tool (IBAT) can run a ‘freshwater report’ for areas of interest which uses IUCN Red List data...”

We have rephrased this text as suggested.

Line 1537. Add some text here on the logic behind Extended Data Table 1 etc, e.g. “We calculated the proportion of species impacted by each threat type for extant and extinct species in two ways: firstly treating CR(PE) and CR(PEW) species as extant (because their extinction has not been confirmed), and secondly treating them as extinct (because this is suspected to be their true status)” – or something along these lines.

We have expanded the legends of Extended Data Tables 2 and 4 to make the logic clearer on the two placements of CR(PE) and CR(PEW) species in these analyses.

Responses to referee comments

Referee #3:

The authors have done a commendable job addressing my primary comments, particularly around improving the clarity of representation of the study's methods and potential limitations which would otherwise be limited by the format used in this journal. I am satisfied with the revisions conducted in response to my prior comments and consider the manuscript greatly improved. I have no further comments and recommend acceptance at this stage.

We are pleased that the referee is satisfied with our edits and thank them again for their thoughtful and thorough review of the manuscript.

Referee #3 in response to edits originally suggested by referee #2:

...they also felt that your discussion of the omission of molluscs needed to go further, in particular to discuss the likely outcomes of their omission from the assessments. For example, does the omission likely result in an underestimate or overestimate of global threat? Please ensure that in your revised text, when general statements or conclusions are made, you should include a qualification of what information may be missing or excluded, and include a justification of this decision and acknowledgment of the implications for the study's conclusions.

We thank referee #3 for reviewing our responses to referee #2's original comments. We have expanded the main text to include further discussion of the implications of the omission of freshwater molluscs on our analyses, notably comparing results on extinction risk (threatened, Extinct, and Data Deficient species), threats and habitats from our analyses and from a previous analysis of the sampled Red List for molluscs (Bohm et al. 2021).

Questions from the Editor:

Does the manuscript have flaws which should prohibit its publication?

The articles' objectives, results, and their implications must be more clearly and accurately described (see detailed comments below).

Do you feel that the results presented are of immediate interest to many people in your own discipline, or to people from several disciplines?

Yes, the results are of general interest to professionals in my field and other environmental fields. The loss of global biodiversity *should* be of interest to a general audience and this type of study and the associated messaging deserve much greater attention from the scientific community at large.

Overall Comments

The manuscript reflects a tremendous amount of long-term effort by more than a thousand scientists bringing together data of mixed quality and engaging in a well-established expert solicitation process. The conservation issues covered, notably the freshwater biodiversity crisis and its severity across space, are paramount for environmental and sustainability sciences. I feel that the broader objectives of the paper should be more clearly stated, especially early on. In addition, more broad/general references to the approach used and at least the conceptual basis of the methods employed should be more clear outside of the methods section. For an interdisciplinary audience reading a paper formatted with the methods section last, the authors must be realistic that most readers will not see the methods section narrative. Accordingly they should be more clear about the nature of the investigation (in particular the nature of the quantitative data used, and the degree of reliance on expert opinion) and more carefully phrase, interpret, and qualify results in the discussion section. I think substantial revisions to the text will be necessary to accomplish these goals but the paper is absolutely worthy of publication and the research is of substantial scientific merit for the broader environmental conservation community. The title is also vague uninformative as to the study's findings.

Comments on Text

Bold First Paragraph

28-29 – How can the authors reconcile this statement with recent studies on population declines in freshwater megafauna, e.g., He et al., 2019? Is the idea that this was merely an analysis of trends and not extinction risk per se? If so then the authors should specify early on what is meant by 'extinction risk' in this context. As currently used with little qualification (here or in the main text), it implies that the IUCN standard methodologies are the *de-facto* definition of extinction risk, which is a bit misleading for non-experts. In the main text especially, a thorough discussion of the advantages and disadvantages/limitations of the IUCN classification system is worthwhile.

In this paragraph, at the very least, authors should specify that they are specifically referencing the IUCN definition of risk and explain that it is a semi-quantitative rating system.

The authors also seem to use the term somewhat interchangeably with the umbrella term “biodiversity assessment”, and this distinction or shared definition should be clarified.

36 – I think the word “reveals” is a bit heavy-handed here. There is no doubt that freshwater biodiversity is in crisis, but my understanding of the analyses conducted is that they *suggest* that this proportion of species are threatened with extinction. Actual quantitative assessments of population trends and demographic rates were not undertaken. The authors must find an appropriate balance between promoting the impressive, laudable scale of the research undertaken, the gravity of the findings, and the limitations of the analytical approaches. This is especially true considering that the methods are relegated to the end of the manuscript in this format and a small proportion of readers will realistically ever look through them.

Main Text

79 – it is around this point that the authors should clarify specifics of the approach for IUCN risk assessment/biodiversity assessment; this should be considered and treated as important background information and separate from / less detailed than the methods. The implications, limitations, and caveats of these assessments should also be described.

It should be clear to readers that assessments are undertaken by convening panels for a synthesis of expert opinion and not purely quantitative analyses.

129 – Although the authors briefly touch upon the point, I think it should be discussed whether the high number of documented extinctions in the United States is an artifact to better data availability.

163-167 – This reported result is described with a much greater degree of interpretation and support than many preceding points, and cites more literature explaining details about why/how this particular threat is likely so prominent. Other described results like the preceding ones pertaining to impacts on decapods and odonates should be treated with similar attention. Alternatively if such information on the nature of these threats is not well understood, it should be clearly stated by the authors. At present there is no clear reason for this differential treatment of findings.

171-179 – What can the authors say about the implications of this discrepancy? Does it imply anything about changing pressures on freshwater systems, or a filter effect on biota across time?

191 – the geographic spread/habitat specialization of these threatened odonates might be worth adding here for context. It is not intuitively clear why forest loss would be such an important driver of their declines. Is it from watershed impacts of deforestation or do they actually occupy

forest habitats? Are they found in the actual forests or in wetland habitats embedded in the forests?

Are wetland habitats classed as “Forest” swamps? Streams running through forests? Mangroves?

233 – the authors may want to comment on and perhaps provide a threshold on what they consider a “good” surrogate here. Or, they could provide ranges of quality for surrogates based on SAI value. For example, it sounds like anything below 0.40 is considered “poor”, while values below 0.5 and above 0.4 are considered “reasonable”, etc.

254 – Are any of these values sufficiently negative to be useful? I.e., if the relationship is worse than random, can it be informative in the inverse? This should be clarified.

Discussion

259-261 – Again, I think the authors should be clear about what mechanistically is meant by “assessing extinction risk” here. Is it quantitative? Semiquantitative? This will not be common knowledge to a broad audience. That is, the authors should be more clear about the nature of data collection and assessment in the study, and not treat it as some sort of definitive analysis. I certainly don’t disagree with the importance of the findings, nor do I question the immense effort of the undertaking, but language is needed to clarify that this is largely the result of expert consultation, so findings represent a strong scientific consensus (which is nevertheless valuable).

Phrases like “Although agriculture and invasive species are major threats...” are in their phrasing overstretched. At least given the understanding conveyed by the methods (and if this is not true, then the authors should substantially clarify the methods), it would be more appropriate to describe study findings as “Although agriculture and invasive species *are widely understood [or considered] to be* major threats...” etc.

274 – Missing a recent key citation on the potential applications of NbS to the freshwater biodiversity crisis <https://doi.org/10.1371/journal.pwat.0000126>

268-271 – There are many papers from the last 1-3 years on this topic, and some merit acknowledgment here. The better integration of ecological research, biodiversity conservation, and water resources management has been repeatedly proposed in the Water Diplomacy / IWRM literature (ecological stakeholder analog framework). Vollmer and others have advocated for watershed health as a way of combining broader water management and environmental goals under an integrated, local-scale decision-making lens.

277 – This is a revealing sentence that highlights what should be clarified in the definition of “extinction risk” necessary in earlier segments of the paper. Given Nature’s format of having methods at the end of the paper, at least some brief explanation of this term, clarifying that it is not a quantitative viability analysis, is thus important early in this manuscript.

282 – Is restricted range *relative to historical baselines* considered in this context? Could it be more or less helpful in assessing freshwater species which naturally have small ranges?

292 – citizen science and other participatory monitoring at local scales warrant mention here as a potential stopgap for the high need and low available funding for monitoring.

The authors should cite some examples of what “improved regulation” should look like, as well as involvement of stakeholders. There is a growing literature of work on stakeholder involvement in freshwater resources management and biodiversity conservation.

312 etc. – Since the broader focus of this paper is on spatial representation and areal coverage, there should be some discussion of caveats, e.g., nonuniform distribution of organisms within these ranges, disproportionate importance of certain areas/habitat types within ranges for life history / population dynamics, and so on.

313 – the poor alignment of water stress and freshwater biodiversity threat status is too large a result to warrant such short discussion. Why do the authors think that this integrated measure, even if abiotic, does such a poor job? What biologically important factors is it missing?

327 – what is “cross-sectorial” meant to signify? Which additional sectors should be brought in, and why? There is recent literature for example on overlaps between biodiversity conservation and water resources management, agriculture, infrastructure, and so on.

336 – literature on relational value of biodiversity and ecosystems should be cited here. Another great reference not to be omitted is Cooke et al (2021) freshwater ethic - <https://doi.org/10.1002/aqc.3537>

347 – how long is “a while”?

354 – The narrative of this paper seems to primarily circulate around protected areas as the major conservation action of concern. I understand if this is the M.O. of IUCN, but it is worth some discussion of other conservation strategies like traditional agriculture, NbS, invasive species removal, and so on, as the protected area paradigm is coming under heavy criticism for its social equity implications in recent years. That is also worth specific mention for non-expert readership.

Figures

Y-axis label for Figure 1; The quantity “Species per red list category” cannot have the unit (%); I think what the authors mean is Proportion of species per red list category.

Figure 2 & 3- could this information be more effectively displayed as a table or as a grid with color-coded cells to highlight results of particular importance? This number of bars across the X axis seems unwieldy. Alternatively, perhaps authors could consider using arrows or boxes to highlight the key take-homes points within these figures.

Figure 3 – Discussion of the findings of this analysis outside of the methods section should clarify whether species occupying multiple habitat types were “double counted”. The authors should be cautious of the potential for readers to infer from these results that certain habitat types are more worthy of conservation than others simply on the basis of (threatened) species richness metrics.

Methods

623 – What did these tabular and spatial data consist of? Occurrences? Counts?

679-80 – What were the thematic classes used for each of these categories and how were they assigned?

881 – some brief (1-2 sentence) explanation of the function and statistical mechanism of SAIs would be helpful here. The formula is described below, but the interpretation and utility should be discussed.

933 – This is worth mentioning in the discussion paragraph around the results of water stress. Were there indeed no freshwater species in these areas, or are they already missing due to a filter effect?